# TIGHTER PRIVACY AUDITING OF DP-SGD IN THE HIDDEN STATE THREAT MODEL

**Tudor Cebere**[*]
Inria, Université de Montpellier
`tudor.cebere@inria.fr`

**Aurélien Bellet**
Inria, Université de Montpellier
`aurelien.bellet@inria.fr`

**Nicolas Papernot**
University of Toronto & Vector Institute
`nicolas.papernot@utoronto.ca`

## ABSTRACT

Machine learning models can be trained with formal privacy guarantees via differentially private optimizers such as DP-SGD. In this work, we focus on a threat model where the adversary has access only to the final model, with no visibility into intermediate updates. In the literature, this "hidden state" threat model exhibits a significant gap between the lower bound from empirical privacy auditing and the theoretical upper bound provided by privacy accounting. To challenge this gap, we propose to audit this threat model with adversaries that *craft a gradient sequence* designed to maximize the privacy loss of the final model without relying on intermediate updates. Our experiments show that this approach consistently outperforms previous attempts at auditing the hidden state model. Furthermore, our results advance the understanding of achievable privacy guarantees within this threat model. Specifically, when the crafted gradient is inserted at every optimization step, we show that concealing the intermediate model updates in DP-SGD does not enhance the privacy guarantees. The situation is more complex when the crafted gradient is not inserted at every step: our auditing lower bound matches the privacy upper bound only for an adversarially-chosen loss landscape and a sufficiently large batch size. This suggests that existing privacy upper bounds can be improved in certain regimes.

## 1 INTRODUCTION

Machine learning models trained with non-private optimizers such as Stochastic Gradient Descent (SGD) have been shown to leak information about the training data (Shokri et al., 2017; Yeom et al., 2018; Balle et al., 2022; Haim et al., 2022). To address this issue, Differential Privacy (DP) (Dwork et al., 2006) has been widely accepted as the standard approach to quantify and mitigate privacy leakage, with DP-SGD (Abadi et al., 2016) as the defacto algorithm to train machine learning models with DP guarantees. DP-SGD follows the same steps as standard SGD but clips the gradients' norm to a threshold before perturbing them with carefully calibrated Gaussian noise, providing differential privacy guarantees for each gradient step.

As the practical importance of differential privacy grew, the need to track the *privacy loss* efficiently across the entire training process (*privacy accounting*) became critical: indeed, optimal utility is obtained by adding the minimum amount of noise required to achieve the desired privacy guarantee. Existing privacy accounting techniques rely on privacy composition (Kairouz et al., 2015; Abadi et al., 2016; Gopi et al., 2021; Doroshenko et al., 2022) to derive the overall privacy guarantees of a training run of DP-SGD from the guarantees of each gradient step. Privacy amplification techniques (Kasiviswanathan et al., 2011; Feldman et al., 2018; Erlingsson et al., 2019; Cyffers and Bellet, 2022; Altschuler and Talwar, 2022) can further decrease the overall privacy loss by exploiting the non-disclosure of certain intermediate computations. For instance, privacy amplification by subsampling relies on the secrecy of the randomness used to select the mini-batches.

---

[*]Part of this work was done while visiting University of Toronto & Vector Institute

Despite recent improvements in privacy accounting, training over-parameterized neural networks from scratch with differential privacy typically results in either weak privacy guarantees or significant utility loss (Tramèr and Boneh, 2021). While a possible explanation could be that the privacy accounting of DP-SGD based on composition is overly conservative, (Nasr et al., 2021; 2023) refuted this hypothesis using *privacy auditing*. Leveraging the fact that differential privacy upper bounds the success rate of any adversary that seeks to infer private information from the output of DP-SGD, they showed that it is possible to instantiate adversaries that achieve the maximal success rate allowed by the privacy accounting upper bound. This negative result suggests that the only hope to improve the privacy-utility trade-offs of DP-SGD is to *relax the underlying threat model*, i.e., to make additional assumptions limiting the adversary's capabilities. We focus on one capability granted to the adversary in prior work, namely that all intermediate models (i.e., training checkpoints) are released.

In this work, we consider a natural relaxation where intermediate models are concealed and only the final model is released. This threat model, often referred to as *hidden state*, is particularly relevant in practice, encompassing scenarios such as open-sourcing a trained model by publishing its weights. From a theoretical perspective, recent work has demonstrated that the hidden state model can yield significantly improved privacy upper bounds for DP-SGD compared to those derived through standard composition (Ye and Shokri, 2022; Altschuler and Talwar, 2022). This improvement can be attributed to the phenomenon of *privacy amplification by iteration* (Feldman et al., 2018; Balle et al., 2019): in a nutshell, the privacy of a data point used at earlier stages of the optimization process improves as subsequent steps are performed. However, these results only hold for convex problems, leaving a pivotal question unanswered: *Does the privacy of non-convex machine learning problems improve when intermediate models are withheld?* Empirical lower bounds obtained through privacy auditing (Nasr et al., 2021; 2023; Steinke et al., 2023) suggest that indeed, concealing intermediary model amplifies privacy, exposing a gap between the empirical lower bound and the theoretical upper bound in the hidden state threat model. However, it remains unclear whether this gap arises from genuine privacy amplification (i.e., a loose upper bound) or from the suboptimality of existing adversaries (i.e., a loose lower bound). This raises another crucial question: *How can one design worst-case adversaries when the intermediate models are concealed?*

**Our contributions.** We adapt the *gradient-crafting* adversarial approach of Nasr et al. (2021) to align with the hidden state model, leveraging the key observation that all privacy accounting techniques for DP-SGD are designed to protect against worst-case *gradients*. Instead of crafting a data point (*canary*) that gets added to the training set as in prior attempts to audit the hidden state model (Nasr et al., 2023; Steinke et al., 2023), our adversaries craft a *sequence of gradients* prior to the execution of the algorithm (i.e., without knowledge of the intermediate models). The crafted gradients are then added to gradients computed on real training points to yield the highest possible privacy loss for the final model. In other words, our adversaries abstract away the canary and directly decide the gradient it would have produced when inserted at a given step of DP-SGD. But how do we pick the gradient sequence leading to the worst-case leakage?

In the scenario where crafted gradients are inserted at every iteration of DP-SGD, we demonstrate that gradient-crafting adversaries which allocate the maximum magnitude permitted by DP-SGD to a single gradient dimension are optimal: they imply privacy lower bounds that match the known upper bounds given by the privacy accountant of Gopi et al. (2021). Therefore, *our results reveal that releasing only the final model does not amplify privacy in this regime*. In the case of small models, we achieve these tight privacy auditing results by carefully selecting the gradient dimension, whereas for over-parameterized models, the dimension selection can be arbitrary.

When the crafted gradient is *not* inserted at every step, we find that the above adversaries still outperform canary-crafting adversaries by a significant margin but cannot reach the privacy upper bounds. We show that part of the gap can be attributed to the inability of our adversaries to influence the gradients on real training data in steps where the crafted gradient is not inserted. To address this gap, we design an adversary that crafts both the gradient and the loss landscape. Our results uncover two distinct regimes: (i) when the batch size is large compared to the noise variance, our adversary proves to be optimal, yielding a privacy lower bound that matches the known upper bound and thus demonstrating the absence of privacy amplification; and (ii) when the batch is small relative to the noise variance, we show strong evidence of privacy amplification for non-convex problems, although the effect is qualitatively weaker than in the convex case. Our findings advance the understanding of privacy guarantees in the hidden state model, and lay the groundwork for the design of better privacy accounting techniques for this threat model.

## 2 BACKGROUND

### 2.1 DIFFERENTIAL PRIVACY AND DP-SGD

Differential Privacy (DP) has become the de-facto standard in privacy-preserving machine learning thanks to the robustness of its guarantees, its desirable behaviour under post-processing and composition, and its extensive algorithmic framework. We recall the definition below and refer to Dwork and Roth (2014) for more details. Here and throughout, we denote by $\mathcal{D}$ the space of datasets.

**Definition 1** (($\varepsilon, \delta$)-Differential Privacy). *A randomized mechanism $\mathcal{M}$ is $(\varepsilon, \delta)$-DP if for all neighboring datasets $D \in \mathcal{D}$ and $D' = D \cup \{x\} \in \mathcal{D}$ and for all events $\mathcal{O}$:*

$$P[\mathcal{M}(D) \in \mathcal{O}] \leq e^{\varepsilon} P[\mathcal{M}(D') \in \mathcal{O}] + \delta. \tag{1}$$

In the above definition, $\delta \in (0, 1)$ can be thought of as a very small failure probability, while $\varepsilon > 0$ is the privacy loss (the smaller, the stronger the privacy guarantees).

The workhorse of private machine learning is the Differentially Privacy Stochastic Gradient Descent (DP-SGD) algorithm (Song et al., 2013; Bassily et al., 2014; Abadi et al., 2016). Let $D$ be the training dataset, $\theta$ the model parameters and consider the standard empirical risk minimization objective $\min_\theta \frac{1}{|D|} \sum_{x \in D} \ell(\theta; x)$ where $\ell$ is a loss function differentiable in its first parameter. DP-SGD follows similar steps as standard SGD but ensures differential privacy by (i) bounding the contribution of each data point to the gradient using clipping and (ii) adding Gaussian noise to the clipped gradients. Formally, starting from some initialization $\theta_0$, DP-SGD performs $T$ iterative updates of the form:

$$\theta_{t+1} = \theta_t - \tfrac{\eta}{|B_t|}\left( \sum_{x \in B_t} \text{clip}(\nabla_{\theta_t} \ell(\theta_t; x), C) + Z_t \right), \tag{2}$$

where $\eta > 0$ is the learning rate, $B_t \subseteq D$ is a mini-batch of data points, $\text{clip}(g, C) = g \cdot \min(1, C/\|g\|_2)$ with $C > 0$ the clipping threshold, and $Z_t \sim \mathcal{N}(0, \sigma^2 C^2 \mathbb{I})$.

### 2.2 THREAT MODELS FOR DP-SGD

DP protects against an adversary that observes the output of either $\mathcal{M}(D)$ or $\mathcal{M}(D')$ and seeks to predict whether the dataset was $D$ or $D' = D \cup \{x\}$, i.e., whether $x$ was included in the input dataset (a.k.a., a *membership inference attack*) (Homer et al., 2008; Shokri et al., 2017). In this context, the threat model specifies which information is observable/known by the adversary. For DP-SGD, in addition to the final model $\theta_T$, threat models typically consider that the adversary has access to (and potentially controls) the model architecture, the loss $\ell$, the initialization $\theta_0$, the dataset (up to the presence of $x$) and differ in which internal states of DP-SGD are observable by the adversary. Below, we recall the two threat models relevant to our work.

**Standard threat model.** Standard privacy accounting techniques analyze DP-SGD as a composition of (potentially subsampled) Gaussian mechanisms (Abadi et al., 2016; Gopi et al., 2021; Doroshenko et al., 2022). Composition allows the adversary to *observe all intermediate models $\theta_1, ..., \theta_{T-1}$* (in addition to $\theta_0$ and $\theta_T$). Previous work has shown that existing privacy accounting techniques are tight in this threat model (Nasr et al., 2021; 2023). However, revealing intermediate models is often unnecessary in practical deployment scenarios (particularly in centralized settings where a single entity conducts the training process) and may degrade the privacy-utility trade-off, thereby motivating the study of the hidden state threat model.

**Hidden state threat model.** Our work studies the scenario where intermediate models $\theta_1, ..., \theta_{T-1}$ are concealed from the adversary, who observes only the final model $\theta_T$. This threat model has attracted much attention recently, with theoretical work proving better privacy upper bounds than in the standard threat model in some regimes (Feldman et al., 2018; Balle et al., 2019; Ye and Shokri, 2022; Altschuler and Talwar, 2022). A fundamental phenomenon underlying these results is "privacy amplification by iteration" (Feldman et al., 2018), which shows that repeatedly applying noisy contractive iterations enhances the privacy guarantees of data used in earlier steps. Unfortunately, applying this general result to DP-SGD is only possible for convex problems, ruling out deep neural networks. The existence of privacy amplification for non-convex problems in this threat model is an open problem (Altschuler and Talwar, 2022) that we study in this work through the lens of privacy auditing.

## 2.3 AUDITING DIFFERENTIAL PRIVACY

Privacy accounting only provides upper bounds on the privacy loss captured by the DP parameters $(\varepsilon, \delta)$, and these bounds are not always tight. Leveraging the relation between DP and the performance of membership inference attacks, privacy auditing aims to produce *lower bounds* on the DP parameters by instantiating concrete adversaries permitted within a given threat model (Jagielski et al., 2020; Nasr et al., 2021). When these lower bounds match the upper bounds given by privacy accounting, we can conclude that the privacy analysis is tight; when they do not, it is possible to improve the privacy accounting and/or the attacks. A privacy auditing pipeline can be broken down into two components: an adversary and an auditing scheme. The high-level process is shown in Algorithm 1.

**Adversary.** Auditing a mechanism $\mathcal{M}$ first requires the design of an adversary $\mathcal{A}$, which seeks to predict the presence or absence of a *canary* point $x^*$ from the information available in the considered threat model. Typically, an adversary $\mathcal{A}$ consists of two subroutines `RankSample` and `RejectionRule`. `RankSample` gives a confidence score that the observed output was generated by sampling from $\mathcal{M}(D)$ rather than from $\mathcal{M}(D \cup \{x^*\})$, which can interpreted as the two hypotheses of a binary test. `RejectionRule` applies a threshold to the confidence scores generated across multiple random runs of $\mathcal{M}$ to decide when to reject the first hypothesis. It then compares these decisions to the ground truth to produce binary test statistics: True Negatives (TN), True Positives (TP), False Negatives (FN), and False Positives (FP).

**Auditing scheme.** The auditing scheme takes as input the hypothesis testing statistics of an adversary and a privacy parameter $\delta$, and outputs a high-probability lower bound $\hat{\varepsilon}$ on the privacy loss of the mechanism $\mathcal{M}$. It comprises two subroutines: `ConfInterval` and `ConvertToDP`. `ConfInterval` converts the adversary statistics to high probability lower bounds for the False Negative Rates $\alpha$ and False Positive Rates $\beta$. `ConvertToDP` then converts these lower bounds into $\hat{\varepsilon}$ for the specified $\delta$ by leveraging the fact that DP implies an upper bound on any adversary's performance. Our experiments will rely on the recently proposed auditing scheme based on Gaussian DP (Nasr et al., 2023), which we describe for completeness in Appendix A.

---

**Algorithm 1** Privacy auditing

**Input:** Audited mechanism $\mathcal{M}$, adversary $\mathcal{A}$, dataset $D$, canary point $x^*$, number of auditing runs $R$, privacy parameter $\delta$
$S \leftarrow [], b \leftarrow []$
$D_0 \leftarrow D$
$D_1 \leftarrow D \cup \{x^*\}$
**for** $i = 1$ **to** $R$ **do**
  $b_i \leftarrow \text{Ber}(1/2)$      ▷ draw a random bit
  $S_i \leftarrow \mathcal{A}.\text{RankSample}(\mathcal{M}(D_{b_i}), D_0, D_1)$
**end for**
TN, TP, FN, FP $\leftarrow \mathcal{A}.\text{RejectionRule}(S, b)$

$\alpha, \beta \leftarrow \text{ConfInterval}(\text{TN, TP, FN, FP})$
$\hat{\varepsilon} \leftarrow \text{ConvertToDP}(\alpha, \beta, \delta)$
**return** $\hat{\varepsilon}$

---

To obtain the tightest possible $\hat{\varepsilon}$, one should perform auditing using a worst-case canary $x^*$ and a worst-case adversary $\mathcal{A}$. This proves to be especially challenging in the hidden state model as the adversary cannot rely on the knowledge of intermediate models. In this paper, we will obtain tighter lower bounds $\hat{\varepsilon}$ by abstracting away the canary and using adversaries that directly craft gradients.

## 3 RELATED WORK IN DIFFERENTIAL PRIVACY AUDITING

The goal of differential privacy auditing is to create worst-case adversaries that maximally exploit the underlying threat model, even if the information used by the adversary might not be available in practical scenarios.[1] A rich line of work has designed adversaries that can tightly and efficiently audit the privacy of learning algorithms when intermediate models are released (Nasr et al., 2021; Maddock et al., 2023; Nasr et al., 2023; Steinke et al., 2023). The first adversarial construction (the malicious dataset attack in Nasr et al. (2021)) to reach the theoretical upper bound given by privacy accounting for DP-SGD used a restrictive threat model in which the adversary controls the entire learning process, including the dataset, the mini-batch ordering, all hyperparameters *and intermediate models*. In follow-up work by Nasr et al. (2023), a nearly matching lower bound was obtained for a gradient-crafting adversary that does not need to control the dataset, the minibatches or the hyperparameters but *still requires access to intermediate models*.

---

[1]This is in contrast to membership inference attacks seeking feasibility against real systems (Shokri et al., 2017; Yeom et al., 2018; Carlini et al., 2022; Zarifzadeh et al., 2023).

Prior attempts at auditing the hidden state model used adversaries employing a loss-based attack that targets a canary obtained by flipping the label of a genuine data point (Nasr et al., 2021; 2023; Steinke et al., 2023; Annamalai and Cristofaro, 2024). The idea is to generate an outlier from an in-distribution data point so that the loss of the model on this canary is high when it is not part of the training set but drops significantly when used during training. Auditing the hidden state model with these adversaries yields privacy lower bounds that exhibit a significant gap compared to privacy accounting upper bounds for the standard threat model (Nasr et al., 2021; 2023; Steinke et al., 2023; Annamalai and Cristofaro, 2024). This gap has two possible explanations: (i) these adversaries are suboptimal and stronger adversaries exist in the hidden state model; and/or (ii) the upper bounds are not tight. Understanding the reasons for this gap, so as to gain better knowledge of the properties of DP-SGD in the hidden state model, is the main motivation for our work. We demonstrate that gradient-crafting adversaries can provide tighter auditing results, thereby proving the validity of hypothesis (i). We show that in certain regimes, our auditing results match the privacy upper bounds, indicating that releasing intermediate models does not increase the privacy loss. However, we also identify regimes where a gap remains, thus providing compelling evidence in support of hypothesis (ii). Specifically, our findings suggest the presence of a privacy amplification by iteration phenomenon in the non-convex setting, albeit weaker compared to what is theoretically established for the convex case (Feldman et al., 2018; Balle et al., 2019; Bok et al., 2024), but less restrictive than the results of Asoodeh and Diaz (2023). The latter require projecting the iterates onto a convex set of bounded diameter or a strong enough decay term on the parameters, which are unrealistic assumptions in practical scenarios involving non-convex and over-parameterized models like the ones considered in our work.

In Table 1 (Appendix B), we provide a summary of prior auditing results for DP-SGD, highlighting in particular whether the considered adversaries are compatible with the hidden state threat model.

## 4 GRADIENT-CRAFTING ADVERSARIES FOR THE HIDDEN STATE MODEL

In all threat models, DP-SGD enjoys a simplified interpretation as a sequence of $T$ sum queries with bounded terms, where the $t$-th query corresponds to summing the clipped gradients over the mini-batch $B_t$ in Equation 2. From this perspective, privacy accounting techniques for DP-SGD are designed to accommodate any possible gradients with a bounded norm $C$, irrespective of whether they originate from a fixed dataset. This highlights a key limitation of adversaries used so far in the hidden state model: due to the non-convexity of the objective function, any sequence of gradients for the canary $x^*$ is likely to be possible, but *how* to craft a worst-case $x^*$ is unclear (and in fact out of scope for privacy auditing). Instead, we allow the adversary to directly *craft any gradient sequence*, considering that there could be a data point that would generate that sequence. To the best of our knowledge, there is no theoretical restriction on the gradients that neural networks can produce; therefore, we cannot rule out the possibility that some input point could generate an arbitrary sequence of gradients.

**Threat model.** As per the hidden state, only the final model $\theta_T$ is revealed while the intermediate models $\theta_1, \ldots, \theta_{T-1}$ are kept hidden. We assume that the adversary knows the model architecture, the loss function $\ell$, the initialization $\theta_0$, the dataset $D$ and the mini-batches $B_0, \ldots, B_{T-1}$.[2] Considering the mini-batches to be known, as done for instance in Feldman et al. (2018), allows to isolate the impact of concealing the intermediate models from other factors (such as privacy amplification by subsampling). We consider the hyperparameters of DP-SGD ($\eta, C, \sigma$) to be fixed and identical for all optimization steps to avoid uninteresting edge cases (e.g., setting the learning rate $\eta$ to 0 in all optimization steps that do not use the crafted gradients). This requirement can be lifted by switching to Noisy-SGD as in Feldman et al. (2018); Altschuler and Talwar (2022), where hyperparameters like the learning rate or the batch size are part of the privacy definition.

**Gradient-crafting adversaries.** Following the above threat model, we allow the adversary to craft an arbitrary gradient sequence as long as it is not a function of the intermediate models. In other words, the adversary must decide on a sequence of gradients before training starts, in an *offline* way, to audit a *complete, end-to-end training run of DP-SGD*. We stress that this is in stark contrast to Nasr et al. (2023), who use gradient-crafting adversaries to audit each step of DP-SGD and then leverage composition to derive a privacy lower bound for the overall training run. Their approach thus explicitly relies on the adversary having access to all intermediate models, whereas ours does not make this assumption nor use this information in any way.

---

[2]When inserted, the crafted gradient is added to the genuine gradients of the mini-batch.

Our construction helps to understand the hidden state threat model and its properties by decoupling the privacy leakage induced by a worst-case sequence of gradients from the *craftability* of a canary that could produce that sequence. Allowing the adversary to craft a gradient sequence directly circumvents two issues in prior attempts to audit the hidden state model (Nasr et al., 2023; Steinke et al., 2023):

- **Saturating the gradient norm:** The canary point crafted by prior adversaries is not guaranteed to saturate the gradient clipping threshold throughout training. The adversary's performance becomes architecture-dependent: for example, a small convolutional neural network has a small canary norm at the start of the training compared to a ResNet, as observed in Figure 8 of the Appendix. Thus, for a tight audit, one must tune the clipping threshold according to the canary gradient norm, which the adversary cannot access in our threat model.

- **Hypothesis testing:** Prior adversaries need to test the presence of a sequence of gradients they cannot access, so they use the model's loss as a proxy confidence score (`RankSample` in Algorithm 1): a lower loss implies a higher confidence that a sample was used during training. By allowing the adversary to pick the sequence of gradients, we can align the choice of confidence score to the way adversarial information is encoded in the gradients (as will be evident in the adversaries we propose below), thereby achieving superior testing performance.

**Concrete adversary instantiations.** We propose two instantiations of our gradient-crafting adversaries, which we will use to perform privacy auditing in the next section (see ):

1. **Random Biased Dimension** ($\mathcal{A}_{GC}$-R): The adversary picks a random dimension and crafts gradients with magnitude $C$ in this dimension. To test whether crafted gradients were inserted (`RankSample` in Algorithm 1), the adversary uses the difference between $\theta_T$ and $\theta_0$ in that dimension as the confidence score (`RankSample` in Algorithm 1).

2. **Simulated Biased Dimension** ($\mathcal{A}_{GC}$-S): The adversary simulates the training algorithm and picks the least updated dimension. Then, it crafts gradients with magnitude $C$ in that dimension. To test the presence of crafted gradients, the adversary uses the same confidence score as $\mathcal{A}_{GC}$-R.

These two adversaries are designed to be as simple as possible. Other gradient constructions can be used, such as crafting a random gradient following the method proposed by Andrew et al. (2024) (see Figure 7 of the appendix for a comparison with the adversaries we use). However, recent findings on privacy backdoors can further justify our choice of adversaries, where the model architecture and parameters are chosen adversarially to leak certain input points. Specifically, Feng and Tramèr (2024) propose a specific construction of architecture and initialization in which a well-chosen input point generates gradients nearly concentrated in a single dimension throughout training. This demonstrates that in some cases, our adversaries $\mathcal{A}_{GC}$-R and $\mathcal{A}_{GC}$-S can be instantiated by inserting a canary point.

For more details on our two gradient-crafting adversaries, see Appendix C, and refer to Algorithm 4 for the corresponding auditing procedure, adapted from Algorithm 1.

## 5 Privacy Auditing Results on Real Datasets

### 5.1 Experimental Setup

+**Training details.** We perform auditing on two datasets: we choose CIFAR10 (Krizhevsky, 2009) as a representative dataset for the state-of-the-art in differentially private training (Tramèr and Boneh, 2021; De et al., 2022), and Housing (Pace and Barry, 1997) to underline the hardness of auditing smaller models in the hidden state. We use fixed hyperparameters for each dataset: on CIFAR10, the batch size is 128, and the learning rate is 0.01, while on Housing, the batch size is 400, and the learning rate is 0.1. Training is done with DP-SGD with no momentum. We use three models: a fully connected neural network (FCNN) for the Housing dataset (Pace and Barry, 1997), a convolutional neural network (CovNet) (LeCun et al., 1989) and a residual network (ResNet) (He et al., 2016) for CIFAR10. A detailed description of the models can be found in Appendix D.

**Baseline adversary.** As baseline, we adopt an adversary used in prior related research (Nasr et al., 2023; Steinke et al., 2023). This adversary selects a point from the training dataset and flips its label to generate an outlier which serves as the canary $x^*$. The loss of the model on this canary is used as confidence score to test whether it was inserted or not. We refer to this baseline as the "loss-based adversary", denoted by $\mathcal{A}_L$.

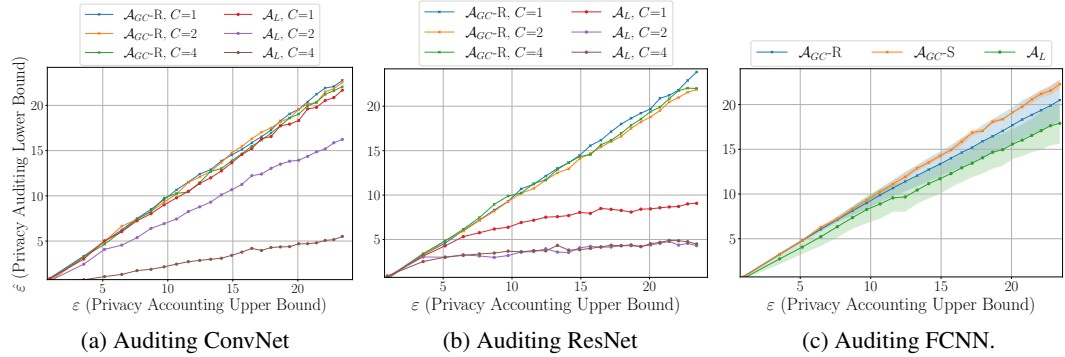

Figure 1: Auditing results for $\mathcal{A}_{GC}$ (ours) and $\mathcal{A}_L$ on ConvNet (Fig. 1a) and ResNets (Fig. 1b) at periodicity $k = 1$ and $C \in \{1, 2, 4\}$. In Fig. 1c we present the results for $\mathcal{A}_{GC}$-R (ours), $\mathcal{A}_{GC}$-S (ours) and $\mathcal{A}_L$ on FCNN (Housing dataset) at periodicity $k = 1$.

**Privacy accounting & auditing.** We compute privacy upper and lower bounds for the crafted gradient (for $\mathcal{A}_{GC}$) or canary ($\mathcal{A}_L$) as follows. The theoretical privacy upper bound is given by the accountant of Gopi et al. (2021). We audit three scenarios where the accountant gives equivalent privacy guarantees for the crafted gradient or canary by inserting it at different periodicity $k \in \{1, 5, 25\}$, accounting only for the steps where the insertion occurs and adjusting the time horizon $T$ accordingly. When $k = 1$, the model is trained for $T = 250$ steps, and the crafted gradient or canary is inserted at every step; when $k = 5$, the model is trained for 1250 steps and insertion occurs every 5 steps; and similarly for $k = 25$. We report both the theoretical and empirical epsilons at a fixed $\delta = 1e^{-5}$, but note that we can generate the complete privacy curve $(\varepsilon, \delta(\varepsilon))$, see Figure 11 of the appendix. For auditing, we rely on the recently proposed scheme based on Gaussian DP (described in Appendix A for completeness) as it allows accurate auditing with a small number of auditing runs (Nasr et al., 2023). Details on training and auditing parameters can be found in Appendix D.

**Remark 1** (On the impact of known initialization). *As explained in our threat model of Section 4, we consider the initialization $\theta_0$ known to the adversary. This is a standard assumption made explicitly or implicitly by virtually all privacy accounting techniques, including those tailored to the hidden state (Feldman et al., 2018). Nevertheless, we show in Figure 12 of the appendix that commonly used random initializations like Kaiming (He et al., 2015) or Xavier (Glorot and Bengio, 2010) do not significantly affect our auditing results, as they only add a bit of variance at initialization.*

**Remark 2** (On pre-trained models). *While we consider here that models are trained from scratch on private data, our techniques and results also apply to the scenario where a model pre-trained on public data defines a new (potentially smaller) model to be fine-tuned on private data. We illustrate this by fine-tuning a pre-trained AlexNet (Krizhevsky et al., 2012) model by training only the last fully connected layer of the classifier on CIFAR10, see Figure 13 of the appendix.*

## 5.2 AUDITING RESULTS FOR PERIODICITY $k = 1$

**Over-parameterized models.** The results in Figure 1 show that our adversary $\mathcal{A}_{GC}$-R achieves tight auditing results in the hidden state model when the crafted gradient is inserted at every step, a result that had been achieved until now only when the adversary could pick a worst-case dataset $D = \{\emptyset\}$ (Nasr et al., 2021) and set the learning rate to 0 in steps where the canary is not inserted, or when the adversary had access to the intermediate models (Nasr et al., 2023). Note that the baseline loss-based adversary $\mathcal{A}_L$ is indeed not tight and even very loose in some regimes.

The fact that a tight audit can be achieved by our simplest adversary $\mathcal{A}_{GC}$-R (random biased dimension) may seem surprising. We provide a high-level explanation of this phenomenon. As we are auditing over-parameterized models, which are highly redundant, we can expect a genuine gradient to assign on average a magnitude of $\mathcal{O}(\frac{C|\text{minibatch}|}{p})$ to each dimension, where $p$ is the number of parameters of the model. As $p$ is typically orders of magnitude larger than the commonly used batch sizes, genuine gradients in a mini-batch contribute a negligible magnitude to any dimension compared

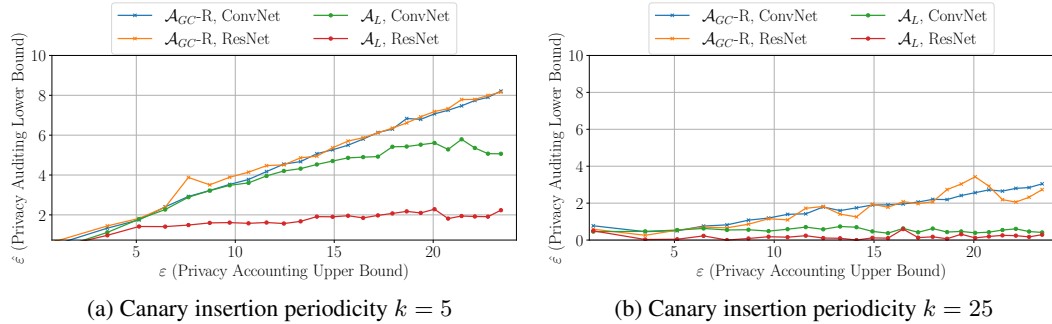

Figure 2: Auditing results for $\mathcal{A}_{GC}$(ours) and $\mathcal{A}_L$ on ConvNet and ResNet (CIFAR10) at privacy parameters $C = 1$ at periodicity $k = 5$ (Figure 2a) and $k = 25$ (Figure 2b).

to the magnitude of $C$ inserted at a single dimension by $\mathcal{A}_{GC}$-R (or $\mathcal{A}_{GC}$-S). Therefore, genuine gradients do not interfere with the contribution of the adversary, regardless of the selected dimension. We see below that this no longer holds when auditing low-dimensional models.

**Low-dimensional models.** Auditing smaller models is more challenging because the ratio between the mini-batch size and the number of parameters is  larger. To evaluate our adversaries in this regime, we switch to the Housing dataset with the FCNN model, which has only 68 parameters. We report the performance of our two adversaries over five  runs in Figure 1c. We observe that randomly selecting a dimension ($\mathcal{A}_{GC}$-R) no longer achieves tight results, although it still outperforms the baseline loss-based adversary. Remarkably, our second adversary, which simulates the training algorithm to select an appropriate dimension ($\mathcal{A}_{GC}$-S), recovers nearly tight results with small variance between runs.

To conclude, the tightness of our adversaries implies a novel negative result.

**Implication 1.** *If a data point is used at every optimization step of DP-SGD, hiding intermediate models does not amplify its privacy guarantees.*

### 5.3 Auditing Results at Periodicity $k > 1$

We now study how our adversaries perform when the crafted gradient is inserted at a higher periodicity $k \in \{5, 25\}$. We observe in Figure 2 that (i) we still outperform the baseline loss-based adversary $\mathcal{A}_L$ by a large margin, but (ii) we no longer match the privacy upper bound, and our adversary becomes weaker as we increase $k$. Intuitively, the latter is due to the accumulation of the noise added on genuine gradients during the $k - 1$ iterations between each crafted gradient insertion. However, privacy accounting (i.e., the upper bound) does not take advantage of this accumulated noise: as *the insertion of a crafted gradient at a given step could bias subsequent genuine gradients* towards a particular direction, the best one can do is to resort to the post-processing property of DP, which ensures that subsequent genuine gradients do not weaken the privacy guarantees. In the next section, we investigate whether it is possible to match the privacy accounting upper bound  in this setting, which amounts to designing an adversary capable of crafting gradients that maximally influence subsequent genuine gradients.

**Remark 3.** *Using auditing via Gaussian DP (Nasr et al., 2023) for $k > 1$ implies approximating the trade-off function of the mechanism with a Gaussian one (see Appendix A for definitions). While this approximation can underestimate the privacy guarantees in certain cases like subsampled or shuffled mechanisms (see Dong et al. (2022); Wang et al. (2024)), in our context it is justified by the Central Limit Theorem (Dong et al., 2022), and has been used before when auditing the hidden state (Nasr et al., 2023). Appendix E (Figure 6a) shows the approximation error is indeed negligible in our case.*

## 6 Towards a Worst-Case Adversary for the Hidden State

In this section, we investigate whether reducing (and potentially closing) the gap with the theoretical upper bound observed in Section 5.3 is possible. For simplicity, we consider the case where a crafted gradient is inserted *only* in the first optimization step, and then $T - 1$ subsequent steps are performed

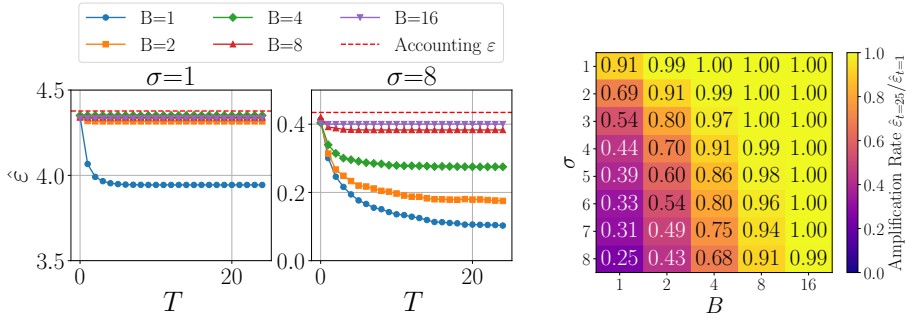

(a) Evolution of the auditing performance across time    (b) Privacy amplification rate

Figure 3: Auditing performance of our adversary $\mathcal{A}_S^{h^*}$ across $T = 25$ steps. Figure 3a shows the evolution of the auditing performance across time for $\sigma \in \{1, 8\}$ and batch size $B \in \{1, 2, 4, 8, 16\}$. Figure 3b gives the privacy amplification rate, i.e., the ratio $\hat{\varepsilon}_{t=25}/\hat{\varepsilon}_{t=1}$ between the privacy auditing lower bounds at step 25 ($\hat{\varepsilon}_{t=25}$) and at step 1 ($\hat{\varepsilon}_{t=1}$).

without inserting crafted gradients. This corresponds to the scenario studied by privacy amplification by iteration (Feldman et al., 2018), but we do not restrict ourselves to convex losses.

We introduce a new class of adversaries $\mathcal{A}_S$ that crafts the gradient *and the loss landscape itself*, thereby controlling the distribution of all $T$ updates. Therefore, $\mathcal{A}_S$ can pick a (non-convex) loss landscape such that the crafted gradient inserted at step $T = 1$ maximally biases subsequent genuine gradients, thereby yielding the highest possible privacy loss for the final model. Note the underlying principle is the same as in Section 4: instead of requiring the adversary to craft a dataset, model architecture and loss function, we allow the adversary to select the loss landscape directly, considering that there could be a dataset that would generate this landscape. As long as the loss landscape is selected before training starts, this is covered by the hidden state threat model we consider.

We formalize the above using the following pair of one-dimensional stochastic processes:

$$\theta_{t+1} = \theta_t - \frac{1}{B}\left(g(\theta_t) + Z_{t+1}\right), \qquad \overline{\theta}_{t+1} = \overline{\theta}_t - \frac{1}{B}\left(g(\overline{\theta}_t) + Z_{t+1}\right), \tag{3}$$

where the first step is either $\theta_1 = Z_1$ (a process that did not use the crafted gradient) or $\overline{\theta}_1 = \nabla^* + Z_1$ (a process that used the crafted gradient $\nabla^*$), with $Z_i \sim \mathcal{N}(0, C^2\sigma^2)$, and $g : \mathbb{R} \to [-BC, BC]$ is a function that abstracts the sum of gradients of the loss on a batch of size $B$ in the DP-SGD update (Equation 2). The goal of the adversary is to choose the function $g$ and the gradient $\nabla^*$, without knowing the intermediate updates, making the distribution of $\theta_T$ and $\overline{\theta}_T$ as "distinguishable" as possible.

**Example 1** (Constant $g$). *Consider the simple case where $g$ outputs a constant $c$, independent of the input. This implies that the Gaussian noise accumulates at every step, and the privacy loss $\varepsilon$ for the last iterate converges to 0 as $T \to \infty$ at a rate of $O(1/\sqrt{T})$.*

As illustrated by Example 1, it is easy to design a function $g$ that amplifies privacy over time. However, the converse objective, i.e., designing a function that provides no or minimal privacy amplification for the final step $T$, appears much more challenging.

**A worst-case proposal.** We propose a concrete adversary $\mathcal{A}_S^{h^*}$ which selects $\nabla^* = C$ and the function $g_{\phi^*}(X) = BC$ if $\phi^*(X) = 1$ and $g_{\phi^*}(X) = -BC$ otherwise, where $\phi^* : \mathbb{R} \to \{0, 1\}$ is the linear threshold function that achieves the minimal False Positive Error Rate $\alpha$ + False Negative Error Rate $\beta$ for distinguishing between $\theta_1$ and $\overline{\theta}_1$ (i.e., testing if the model after 1 step was generated using the crafted gradient $\nabla^*$). More precisely, $\phi^*(X) = 1$ if $X > h^*$ and $\phi^*(X) = 0$ otherwise, with $h^* = \frac{|\nabla^*|}{2} = \frac{C}{2}$ (see Appendix F for a numerical validation). The resulting non-convex loss landscape adheres to the intuition that the crafted gradient maximally biases subsequent steps (see Appendix F).

**Auditing with $\mathcal{A}_S^{h^*}$.** Figure 3a shows the auditing performance of our adversary $\mathcal{A}_S^{h^*}$ for $T \in \{1, \ldots, 25\}$ steps, batch sizes $B \in \{1, 2, 4, 8, 16\}$ and noise variances $\sigma \in \{1, 8\}$ (for more values see Figure 14 in the appendix).[3] Figure 3b gives the ratio between the privacy auditing lower bound

---

[3]As in Section 5, we need to approximate the trade-off function of our mechanism with a Gaussian one (see Remark 3). In practice, the approximation error is again negligible (see Figure 6b in Appendix E).

$\hat{\varepsilon}_{t=25}$ at the last step and the one obtained after the first step, to measure the extent of the privacy amplification phenomenon for various batch sizes $B$ and values of $\sigma$. When this ratio is equal to $1$, there is no amplification and the audit is tight, while values smaller than $1$ indicate the existence of a privacy amplification effect. We observe two general trends: *the amplification rate increases with $\sigma$ and decreases with $B$*. This can be explained as follows. Higher noise variance ($\sigma$) helps forget initial conditions in the stochastic processes (3). Conversely, a larger batch size ($B$) allows subsequent updates to be larger, enabling the initial conditions to propagate better across iterations despite the noise. Indeed, as gradient updates are bounded by $[-BC, BC]$ and the noise remains independent of $B$, increasing $B$ allows the adversary to retain more "signal" about the canary's presence in subsequent gradients.

Unlike the case where the crafted gradient is inserted at every step (see Implication 1), the above results reveal two distinct two regimes, as highlighted in Implications 2 and 3 below. On the one hand, *when the batch size $B$ is large enough relative to the noise variance $\sigma$, our adversary $\mathcal{A}_S^{h^*}$ achieves a tight audit*. In this regime, this adversary is thus optimal and there is no privacy amplification.

**Implication 2.** *For a sufficiently large batch size relative to the noise variance, the privacy accounting of DP-SGD is tight in the hidden state model: hiding intermediate updates does not amplify privacy.*

On the other hand, *privacy amplification seems to occur when the batch size is small enough relative to the noise variance*. As seen from Figure 3a, the privacy loss of $\mathcal{A}_S^{h^*}$ converges to a constant: this is because the privacy loss distributions become sufficiently far away and stop mixing over time. This constant is smaller than the upper bound from privacy accounting, but remains positive, indicating that this potential amplification must be qualitatively weaker than in the convex case.

**Implication 3.** *In the non-convex regime, for a sufficiently small batch size relative to the noise variance, hiding intermediate updates may amplify the privacy guarantees of DP-SGD. However, the privacy loss of a sample does not go to $0$ as subsequent steps are performed, in contrast to the convex case where the privacy loss decreases in $O(1/\sqrt{T})$ (Feldman et al., 2018).*

**Remark 4** (Comparison to parallel work)**.** *Implication 2 was independently established in a recent concurrent work by Annamalai (2024). Their worst-case construction differs from ours, and in particular implicitly assumes that the batch size is large relative to the noise variance. As a consequence, their results overlook the fact that privacy amplification can occur under small batch sizes, which is a practically important regime where stronger privacy guarantees may be achievable.*

## 7 DISCUSSION AND FUTURE WORK

Our work allows tighter auditing of DP-SGD in the hidden state model, and our results yield several implications that advance the understanding of this threat model. For instance, Implication 3 means that the property of converging privacy loss of DP-SGD as $T \to +\infty$ recently established by Altschuler and Talwar (2022) in the convex case cannot hold for non-convex models without additional constraints. Beyond this negative result, our work suggests that a (weaker) form of privacy amplification does occur for non-convex problems when the batch size is small relative to the noise variance.

Beyond differential privacy, our results have consequences for machine unlearning (Bourtoule et al., 2021), which aims to remove a specific data point from a trained model as if it had never been used for training. A popular definition for unlearning is inspired by DP (Guo et al., 2020; Neel et al., 2021; Gupta et al., 2021), with recent work relying on noisy training and privacy amplification by iteration to provide unlearning guarantees (Chien et al., 2024). Implication 3 shows that this approach cannot completely unlearn a data point when considering non-convex models like neural networks.

An intriguing open question arises from the gap between our empirical results on real datasets (Section 5) and the adversary crafting the loss landscape (Section 6) when the canary is not inserted at every step. While recent work on privacy backdoors shows that certain architectures can be manipulated to make a specific data point produce the desired gradient sequence throughout training (Feng and Tramèr, 2024), it seems unlikely that this point could also maximally bias subsequent gradients of genuine training points in the same way as our worst-case adversary $\mathcal{A}_S^{h^*}$. Investigating the extent to which this is feasible in commonly used architectures would enhance our understanding of privacy leakage in deep learning, and could ultimately lead to better privacy accounting for the hidden state model.

ACKNOWLEDGEMENTS

We thank Pierre Stock and Alexandre Sablayrolles for numerous helpful discussions and feedback in the early stages of this project and Santiago Zanella-Béguelin for observing the need to fix the hyperparameters for DP-SGD in our threat model. We want to thank members of the CleverHans Lab for their feedback.

The work of Tudor Cebere and Aurélien Bellet is supported by grant ANR-20-CE23-0015 (Project PRIDE) and the ANR 22-PECY-0002 IPOP (Interdisciplinary Project on Privacy) project of the Cybersecurity PEPR. This work was performed using HPC resources from GENCI–IDRIS (Grant 2023-AD011014018R1).

The work of Nicolas Papernot and Tudor Cebere (while visiting the University of Toronto and Vector Institute) is supported by Amazon, Apple, CIFAR through the Canada CIFAR AI Chair, DARPA through the GARD project, Intel, Meta, NSERC through the Discovery Grant, the Ontario Early Researcher Award, and the Sloan Foundation.

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

# A    DETAILS ON AUDITING VIA GAUSSIAN DP

For completeness, in this section we present the privacy auditing approach proposed by Nasr et al. (2023), which we use in our experiments. We start by introducing some necessary concepts on $f$-DP and Gaussian DP in Section A.1. Then, in Section A.2, we present an overview of how auditing is performed and how all the elements of the general auditing pipeline (Algorithm 1) are implemented.

## A.1    $f$-DIFFERENTIAL PRIVACY AND GAUSSIAN DIFFERENTIAL PRIVACY

$f$-DP and Gaussian DP (Dong et al., 2022) are based on a characterization of DP as binary hypothesis testing. We recall the main concepts below.

**Definition 2** (Error rates). *Let $P$ and $Q$ be two arbitrary distributions and $O$ a sample drawn from either $P$ or $Q$. Let a binary hypothesis test be defined by the following two hypotheses: $H_0$ : "the output $O$ is drawn from $P$" or "$H_1$: the output $O$ is drawn from $Q$". Consider a rejection rule $\phi : \mathcal{R}^d \to [0,1]$ that outputs the probability that we should reject $H_0$. We define the Type I (or false positive) error rate of this rejection rule $\phi$ as $\alpha_\phi \triangleq \underset{P}{\mathbb{E}}[\phi]$ and the Type II (or false negative) error rate as $\beta_\phi \triangleq 1 - \underset{Q}{\mathbb{E}}[\phi]$.*

**Definition 3** (Trade-off functions). *Let $P$ and $Q$ be two arbitrary distributions. We define the trade-off function between $P$ and $Q$ as:*

$$T(P,Q)(\alpha) = \inf_\phi \{\beta_\phi : \alpha_\phi \leq \alpha\}. \tag{4}$$

We can now introduce $f$-DP.

**Definition 4** ($f$-Differential Privacy). *Let $f$ be a trade-off function. Then a mechanism $M$ is $f$-Differentially Private ($f$-DP) if for all neighbouring datasets $D$ and $D'$:*

$$T(M(D), M(D')) \geq f. \tag{5}$$

Trade-off functions describe the lowest Type II error rate achievable by any adversary at any Type I error rate . We say that a mechanism $M_1$ is strictly less private than $M_2$ if $T(M_1(D), M_1(D'))(\alpha) \leq T(M_2(D), M_2(D'))(\alpha)$ for all $\alpha \in [0,1]$. Dong et al. (2022) show $(\varepsilon, \delta)$-DP can be formulated as $f$-DP as follows:

$$f_{\varepsilon,\delta}(\alpha) = \max\{0, 1 - \delta e^\varepsilon, e^{-\varepsilon}(1 - \delta)\} \tag{6}$$

Gaussian Differential Privacy (GDP) is a specialized formulation of $f$-DP where $f$ is the trade-off function of two Gaussian distributions.

**Definition 5** (Gaussian Differential Privacy). *Let $\Phi$ be the cumulative distribution function (CDF) of $\mathcal{N}(0,1)$ and $\Phi^{-1}$ its associated quantile function. A mechanism $M$ is $\mu$-GDP if it is $G_\mu$-DP, where*

$$G_\mu(\alpha) \triangleq T(M(D), M(D'))(\alpha) = \Phi(\Phi^{-1}(1 - \alpha) - \mu). \tag{7}$$

The following result is crucial for auditing: it shows that the error rates of an adversary give a lower bound on the GDP guarantees.

**Lemma 1** (Relating error rates to GDP). *Assume that an adversary has error rates $\alpha_M$, $\beta_M$ in the binary test defined by $P = M(D)$ and $Q = M(D')$ where $M$ is a mechanism and $D$ and $D'$ are two neighboring datasets. Then, if $M$ satisfies $\mu$-GDP, then*

$$\mu \geq \Phi^{-1}(1 - \alpha) - \Phi^{-1}(\beta). \tag{8}$$

Finally, we can convert $\mu$-GDP to $(\varepsilon, \delta)$-DP.

**Corollary 1** (From $\mu$-GDP to $(\varepsilon, \delta)$-Differential Privacy.). *If a mechanism $M$ is $\mu$-GDP then it satisfies $(\varepsilon, \delta)$-DP, where:*

$$\delta(\varepsilon) = \Phi(-\frac{\varepsilon}{\mu} + \frac{\mu}{2}) - e^\varepsilon \Phi(-\frac{\varepsilon}{\mu} - \frac{\mu}{2}). \tag{9}$$

## A.2 AUDITING VIA GDP

Auditing a mechanism $M$ via GDP follows the general approach outlined in Algorithm 1, and amounts to instantiating its different primitives:

1. `RankSample`: We first generate $R$ outputs of $M$, half of them sampled from $M(D)$, the rest from $M(D')$, where $D' = D \cup \{x^*\}$. For each output $i \in \{1, \ldots, R\}$, a score $S_i$ is computed to the adversary's confidence that either $D$ or $D'$ were used as input for $M$. The precise computation of this confidence score depends on the considered adversary and is abstracted by the `RankSample` method in Algorithm 1. We refer to Sections 4-5 for details about the `RankSample` method used by the adversaries we consider.

2. `RejectionRule`: Each $S_i$ is then augmented with its associated ground truth label $b_i$, reflecting if a particular sample has been generated from $D$ or $D'$. The adversary selects a classifier (`RejectionRule` in Algorithm 1) that receives the set $S$ as input and needs to classify it against the ground truth label $b$. We stress that DP holds against *any choice* of such a classifier. Commonly used ones are linear threshold classifiers (Nasr et al., 2023). Once a threshold classifier has been selected, we evaluate the error rates of the classifier using $S$ and $b$ as the ground truth labels, evaluating the False Negative (FN), False Positive (FP), True Negative (TN), and True Positive (TP) of the classifier.

3. `ConfInterval`: Using these statistics, we employ the Clopper Pearson (Clopper and Pearson, 1934) confidence intervals over binomial distributions to get a high-probability lower bound on the Type I error rate ($\alpha$) and Type II error rate ($\beta$) for the adversary's performance. We note that other techniques exist to compute such confidence intervals (Zanella-Béguelin et al., 2023; Lu et al., 2022); however, the benefits we observed practically were negligible in our case.

4. `ConvertToDP`: Finally, given access to the lower bounds on the error rates of the adversary, we can compute the lower bound on the privacy loss of a mechanism $M$ in GDP by using Lemma 1, and translate the result to $(\varepsilon, \delta)$-DP using Corollary 1.

We stress that this technique offers a lower bound on the privacy loss, while techniques like privacy accounting (Abadi et al., 2016; Gopi et al., 2021; Doroshenko et al., 2022) or composition (Dwork et al., 2006; Kairouz et al., 2015; Dong et al., 2022) offer upper bounds on the privacy loss. We call a mechanism *tight* if a lower bound offered by auditing match the upper bound

## B SUMMARY OF PRIOR WORK ON PRIVACY AUDITING OF DP-SGD

We provide in Table 1 a list of adversaries used in prior work on privacy auditing of DP-SGD along with their key properties: whether they are compatible with the hidden state, whether they achieved a tight audit, and the type of canary they used.

## C DETAILS ABOUT OUR ADVERSARIES $\mathcal{A}_{GC}$

The algorithmic description of our adversaries $\mathcal{A}_{GC}$-R and $\mathcal{A}_{GC}$-S are given in Algorithms 2-3.

We discuss different strategies to select the biased dimension for our adversary $\mathcal{A}_{GC}$-S. The general idea is to explore how the different dimensions are updated by simulating the training process, as all the required knowledge is available to the adversary. We have considered two types of simulations: (i) Noisy simulation, in which we use the same noise variance as the simulated process, and (ii) Noiseless simulation, in which we run the training algorithm without noise or clipping. Note that for the noiseless simulation, the training algorithm is deterministic, as the initialization and the batches are fixed and known to the adversary, while in the noisy simulation, the stochasticity of DP-SGD is concealed for the adversary. We use four simulations of the training process. We then proceed to choose how to rank the most suitable dimension to bias, for which we have explored two options: (i) Per Step (PS): selecting the dimension that has the lowest gradient norm accumulated over all the training runs or (ii) Final Model (FM): selecting the dimension that is the closest to initialization at the end of the training run. We experiment with these design choices, resulting in 4 dimension selection strategies. The results in Figure 4 show that the Noiseless-PS strategy performs best.

| Paper | Adversary | HS | TA | CT |
|---|---|:---:|:---:|---|
| Jagielski et al. (2020) | Poisoned Backdoor | ✓ | ✗ | Sample |
| Nasr et al. (2021) | API Access | ✓ | ✗ | Sample |
| | Static Adversary | ✓ | ✗ | Sample |
| | Intermediate Poison Attack | ✓ | ✗ | Sample |
| | Adaptive Poisoning Attack | ✗ | ✗ | Sample |
| | Gradient Attack | ✗ | ✗ | Gradient |
| | Malicious Datasets | ✗ | ✓ | Gradient |
| Zanella-Béguelin et al. (2023) | Experiment 1 | ✓ | ✗ | Sample |
| | Experiment 2 | ✓ | ✗ | Sample |
| Maddock et al. (2023) | Algorithm 1 | ✗ | ✗ | Gradient |
| Nasr et al. (2023) | Algorithm 1 | ✓ | ✗ | Sample |
| | Algorithm 2 | ✗ | ✓ | Gradient |
| Andrew et al. (2024) | Algorithm 2 | ✓ | ✗ | Sample or Gradient |
| | Algorithm 3 | ✗ | ✗ | Gradient |
| Steinke et al. (2023) | audit-type=whitebox | ✗ | ✗ | Sample or Gradient |
| | audit-type=blackbox | ✓ | ✗ | Sample |
| Ours | $\mathcal{A}_{GC}$-R, $k = 1$ | ✓ | ✓ | Gradient |
| | $\mathcal{A}_{GC}$-R, $k > 1$ | ✓ | ✗ | Gradient |

Table 1: Summary of adversaries used in prior privacy auditing of DP-SGD. We report whether the adversary is compatible with the hidden state (HS), if the adversary achieves tight audit in the threat model it operates (TA) and the type of canary (CT) used by the adversary (gradient canaries or sample space canaries).

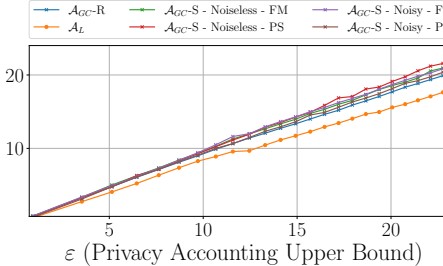

Figure 4: Auditing results of $\mathcal{A}_L$, $\mathcal{A}_{GC}$-R and $\mathcal{A}_{GC}$-S on the Housing dataset. We consider 4 variants of $\mathcal{A}_{GC}$-S, depending on whether the noisy or noiseless simulation is used and whether we rank dimensions based on accumulating per-step updates (PS) or on the final model norm difference (FM).

Figure 5: Auditing results for $\mathcal{A}_{GC}$-S compared to $\mathcal{A}_{GC}$-R at $k \in \{1, 5\}$ on ConvNet and ResNet18. We observe that the two adversaries are equivalent for these over-parameterized models, demonstrating that $\mathcal{A}_{GC}$-S only enhances our attack against under-parameterized models.

In Figure 5, we show that $\mathcal{A}_{GC}$-S using the best strategy is equivalent with $\mathcal{A}_{GC}$-R on our studied over-parameterized models.

---

**Algorithm 2** Gradient Generation for $\mathcal{A}_{GC}$-R (Random Biased Dimension)

---

**Input:** dataset $D$, initialization $\theta_0 \in \mathbb{R}^p$, clipping threshold $C$
$d \leftarrow$ random element from $\{1, \ldots, p\}$
$\nabla^* \leftarrow (0, \ldots, 0) \in \mathbb{R}^p$
$\nabla^*[d] \leftarrow C$
**return** $\nabla^*, d$

---

---

**Algorithm 3** Noisy gradient generation for $\mathcal{A}_{GC}$-S (Simulated Biased Dimension)

---

**Input:** dataset $D$, initialization $\theta_0 \in \mathbb{R}^p$, clipping threshold $C$, noise variance $\sigma^2$, mini-batches
$\quad\quad\{B_1 \ldots B_T\}$, number of runs $r$, parameter selection rank $\in$ {PerStep, FinalModel},
$\quad\quad$ simulation $\in$ {Noisy, Noiseless}

$S = [0]^p$
**for** $j = 1$ **to** $r$ **do**
$\quad$ **for** $i = 1$ **to** $T$ **do**
$\quad\quad Z_t \sim \mathcal{N}(0, C^2\sigma^2\mathbb{I})$
$\quad\quad \theta_{t+1} \leftarrow \theta_t - \frac{\eta}{|B_t|}\Big(\sum_{x \in B_t} \mathrm{clip}\big(\nabla_{\theta_t}\ell(\theta_t; x), C\big)\Big)$

$\quad\quad$ **if** simulation = Noisy **then**
$\quad\quad\quad \theta_{t+1} = \theta_{t+1} + \frac{\eta}{|B_t|}Z$, where $Z \sim \mathcal{N}(0, \sigma^2)$
$\quad\quad$ **end if**

$\quad\quad$ **if** rank = PerStep **then**
$\quad\quad\quad S \leftarrow S + (\theta_i - \theta_{i-1})^2$
$\quad\quad$ **end if**
$\quad$ **end for**

$\quad$ **if** rank = FinalModel **then**
$\quad\quad S \leftarrow |\theta_T - \theta_0|$
$\quad$ **else**
$\quad\quad S \leftarrow \sqrt{S}$
$\quad$ **end if**
**end for**
$d \leftarrow \mathrm{argmin}\, S$
$\nabla^* \leftarrow (0, \ldots, 0) \in \mathbb{R}^p$
$\nabla^*[d] \leftarrow C$
**return** $\nabla^*$, $d$

---

---

**Algorithm 4** Privacy auditing with our gradient-crafting adversaries

---

**Input:** dataset $D$, mini-batches $\{B_1 \ldots B_T\}$, periodicity $k$, number of auditing runs $R$,
        privacy parameter $\delta$, noise variance $\sigma^2$, clipping norm $C$, initial parameters $\theta_0 \in \mathcal{R}^p$,
        standard Gaussian CDF $\Phi$, gradient-crafting adversary $\mathcal{A}_{GC}$

$\nabla^*, d \leftarrow \mathcal{A}_{GC}$            $\triangleright$ get crafted gradient and biased dimension from adversary
$S \leftarrow [], b \leftarrow []$
**for** $j = 1$ **to** $R$ **do**
   $b_i \leftarrow \text{Ber}(\frac{1}{2})$            $\triangleright$ draw a random bit
   **for** $t \in \{1, \cdots, T\}$ **do**

$$\theta_{t+1} = \theta_t - \tfrac{\eta}{|B_t|}\left(\sum_{x_i \in B_t} \text{clip}\left(\nabla_{\theta_t} \ell\left(\theta_t; x\right), C\right) + Z_t\right), Z_t \sim \mathcal{N}(0, \sigma^2)$$

     **if** $t \mod k = 0$   AND   $b_i = 0$ **then**
       $\theta_{t+1} = \theta_{t+1} - \tfrac{\eta}{|B_t|}\nabla^*$            $\triangleright$ Canary is inserted
     **end if**
   **end for**
   $S_i \leftarrow \theta_T[d] - \theta_0[d]$            $\triangleright$ Retrieve the model value at biased dimension $d$
**end for**

$S_{\text{sorted}} \leftarrow \texttt{sort}(S)$
$H \leftarrow \left\{ h_i = \frac{S_{\text{sorted}}[i] + S_{\text{sorted}}[i+1]}{2} \; : \; i = 1, \ldots, |S_{\text{sorted}}| - 1 \right\}$

**for** $h_i \in H$ **do**
   $\hat{b} \leftarrow \{\mathbb{1}\left[S_i \leq h_i\right] \; : \; i = 1, \ldots, |S|\}$
   $\text{TN}, \text{TP}, \text{FN}, \text{FP} \leftarrow \texttt{ConfusionMatrix}(\hat{b}, b)$
   $\alpha_i, \beta_i \leftarrow \texttt{ClopperPearson}(\text{TN}, \text{TP}, \text{FN}, \text{FP})$
   $\mu_i \leftarrow \Phi^{-1}(1 - \alpha_i) - \Phi^{-1}(\beta_i)$
   $\hat{\varepsilon}_i \leftarrow \underset{\varepsilon}{\text{argmin}} \left| \delta - \Phi\left(-\frac{\varepsilon}{\mu_i} + \frac{\mu_i}{2}\right) + e^\varepsilon \Phi\left(-\frac{\varepsilon}{\mu_i} - \frac{\mu_i}{2}\right) \right|$    $\triangleright$ Minimization via binary search
**end for**
**return** $\max\{\hat{\varepsilon}_i : i = 1, \ldots, R\}$

---

## D    EXPERIMENTAL DETAILS ON TRAINING & AUDITING

In our experimental section, we audit three neural network architectures, a ConvNet and a ResNet on the CIFAR10 dataset and a Fully-Connected Neural Network on the Housing dataset. These models were implemented using PyTorch (Paszke et al., 2019), and the DP-SGD optimizer was Opacus (Yousefpour et al., 2021). The hyperparameters for training, privacy guarantees, and privacy auditing parameters for each audited machine learning model are meticulously outlined in Table 2. The ConvNet and ResNet have been audited when trained for 250, 1250, and 6250 optimization steps, totaling approximately 3500 GPU hours.

For clarity, in Algorithm 4 we detail the Gaussian DP-based auditing procedure used for our gradient-crafting adversaries ($\mathcal{A}_{GC}$-R or $\mathcal{A}_{GC}$-S). For each experiment, we audit $R = 5000$ machine learning models using GDP auditing (see Appendix A). For the Clopper Pearson confidence intervals, we use a confidence of $95\%$ (consistent with prior work (Jagielski et al., 2020; Nasr et al., 2021)) to get a lower bound on the Type I error rate ($\alpha$) and Type II error rate ($\beta$) for the adversary's performance. `RejectionRule` is computed by selecting the best linear threshold classifier on $S$. While, in practice, the adversary should pick the classifier on a set of held-out scores $S$, we select the classifier directly on $S$ to save the computational resources required to compute the held-out set $S$. We stress that it is not an issue, as differential privacy guarantees hold against any choice of classifier.

## E    APPROXIMATION ERRORS FOR GDP AUDITING

As explained in Remark 3, using auditing via Gaussian DP (Nasr et al., 2023) (see Appendix A) implies approximating the trade-off function of our mechanism with a Gaussian one (Dong et al.,

| HYPERPARAMETER | FULLY-CONNECTED NN | CONVNET | RESNET18 |
|---|---|---|---|
| LEARNING RATE ($\eta$) | $10^{-2}$ | $10^{-2}$ | $10^{-2}$ |
| BATCH SIZE | 400 | 128 | 128 |
| TRAINABLE PARAMS | 68 | 62006 | 11173962 |
| LOSS FUNCTION | BINARY CROSS ENTROPY | CROSS ENTROPY | CROSS ENTROPY |
| CLIPPING NORM $C$ | 1.0 | { 1.0, 2.0, 4.0 } | {1.0, 2.0, 4.0 } |
| NOISE VARIANCE $\sigma$ | 4 | 4 | 4 |
| FAILURE PROBABILITY $\delta$ | $10^{-5}$ | $10^{-5}$ | $10^{-5}$ |
| AUDITING RUNS | 5000 | 5000 | 5000 |
| CONFIDENCE INTERVAL | 0.95 | 0.95 | 0.95 |

Table 2: Hyperparameters used per model

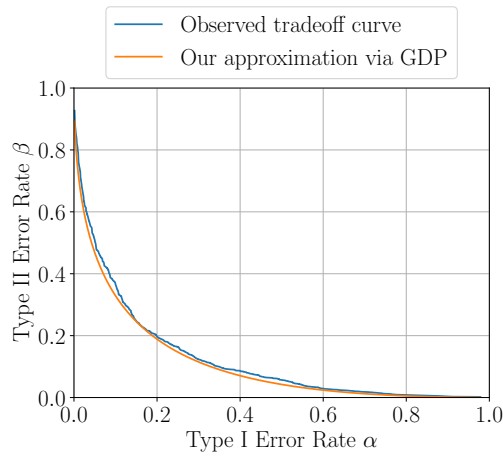

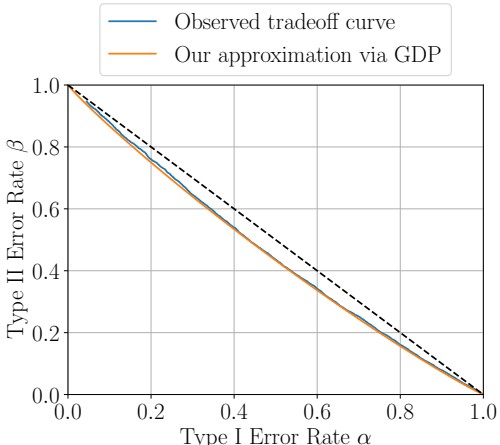

(a) The approximation error of approximating the trade-off function of the observations via GDP for $\mathcal{A}_{GC}$-R when auditing a ConvNet on CIFAR10 when $k = 5$ at step 1250.

(b) The approximation error of approximating the trade-off function of the observations via GDP for $\mathcal{A}_S^{h^*}$ when the $T = 25$ case at privacy parameter $C = 1$ and $\sigma = 4$.

2022). In this section, we present some samples of the privacy trade-off curves that we observe when auditing both $\mathcal{A}_{GC}$ in Figure 6a and $\mathcal{A}_S^{h^*}$ in Figure 6b. For $\mathcal{A}_{GC}$, we show the case where $T = 1250$ when auditing a ConvNet at canary gradient insertion periodicity $k = 5$ on CIFAR10, while for $\mathcal{A}_S^{h^*}$ we present the approximation when $T = 25$. We see that the approximation errors are negligible, and the approximation technique is representative.

This good behaviour is due to the Central Limit Theorem for $f$-DP composition (Dong et al., 2022), which, in a nutshell, states that while individual mechanisms are not well approximated by a Gaussian Mechanism, their composition has a decaying error in the number of compositions performed when approximated via a Gaussian Mechanism.

## F    DETAILS ON OUR ADVERSARY $\mathcal{A}_h^*$

**Visualization of the loss landscape.** Figure 10 shows the loss landscape corresponding to our adversary $\mathcal{A}_h^*$ defined in Section 6. One can see how the crafted gradient can bias subsequent steps. Note that the discontinuity is not essential and can be removed by appropriate scaling.

**Numerical validation of optimality of $h^*$.** In this section, we numerically validate that the choice of threshold $h^*$ in $\mathcal{A}_S^{h^*}$ is indeed optimal for $T = 2$ by comparing it with all possible threshold functions. To search for these functions, we discretize the Type I error rate $\alpha \in [0, 1]$ into $\{\alpha_1 \ldots \alpha_d\}$, and each $\alpha_i$ we look for the rejection rule $\phi_i$ that achieves the lowest Type II error rate $\beta_i$. Given that, in this case, our function $g$ distinguishes between two Gaussian distributions via Neyman-Pearson (Neyman et al., 1933), it is well known that the optimal classifier with a fixed type I error rate is achieved by its

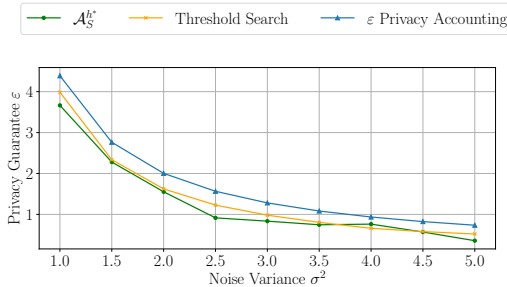

**Algorithm 6** Lower bound search routine over multiple threshold classifiers.

$\{\alpha_1 \dots \alpha_d\} = \text{linspace}(0, \Phi(\frac{C}{2\sigma^2}), d)$
**for** $\alpha_i \in \{\alpha_1 \dots \alpha_d\}$ **do**
$\quad h_i = \Phi^{-1}(1 - \alpha_i)$
$\quad \hat{\varepsilon} = \text{Audit}(g_{\phi_{h_i}})$
**end for**
**return** $\max(\hat{\varepsilon})$

Figure 6: Comparison between the privacy accounting upper bound and the auditing performance of $\mathcal{A}_S^{h^*}$ and our threshold search algorithm when $T = 2$. All results are averaged across five training runs

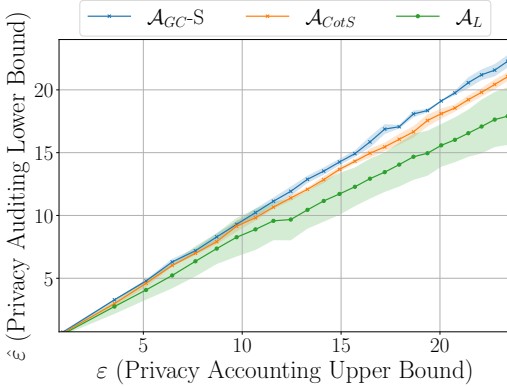

Figure 7: Comparison of privacy auditing performance for $\mathcal{A}_{GC}$-R (ours), $\mathcal{A}_L$ and $\mathcal{A}_{CotS}$ (Andrew et al., 2024) when auditing a FCNN model on the Housing dataset.

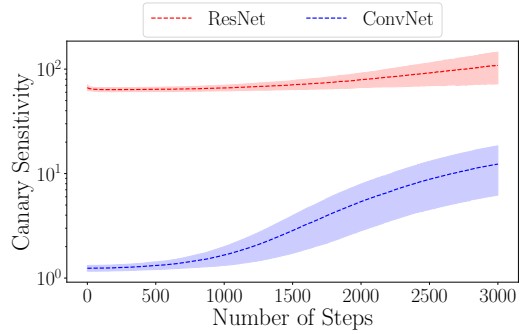

Figure 8: The gradient norm of the canary varies along the optimization path and for different architectures. We report the mean and $\pm 2$ standard deviations across 30 random runs.

associated threshold classifier $h_i = \Phi(1 - \alpha_i)$, where $\Phi$ is the CDF of $\mathcal{N}(0, \sigma^2)$. With $\{h_1 \dots h_d\}$ at hand, we can define multiple $g_{\phi_i}$, and implicitly multiple adversaries $\mathcal{A}_S^{h_i}$ that use $g_{\phi_i}$ as a candidate for $g$ in Eq. 3 which we use to audit, resulting in $\{\hat{\varepsilon}_1 \dots \hat{\varepsilon}_d\}$, out of which we pick the maximum. A detailed description is provided in Algorithm 6.

The results shown in Figure 6 confirm that $\mathcal{A}_S^{h^*}$ employs the optimal threshold classifier to distinguish the input distributions.

# G ADDITIONAL EXPERIMENTAL RESULTS

In this section, we present additional experimental results mentioned in various parts of the main text.

**Alternative gradient-crafting strategy.** The adversaries that we propose in Section 4 and use in Section 5 construct a gradient biased in a single dimension chosen by the adversary. We investigate here an alternative approach inspired by the auditing method of Andrew et al. (2024). In their approach, the crafted gradient is randomly sampled from the unit sphere and then scaled by the sensitivity $C$. For the confidence score (`RankSample` in Algorithm 1), the adversary computes the cosine similarity between the canary gradient and the change in model parameters $\theta_T - \theta_0$, which is compatible with the hidden state model. For completeness, Algorithms 7-8 formally show how the generation of the gradient and the computation of the confidence score is performed. We denote this adversary by $\mathcal{A}_{CotS}$.

---

**Algorithm 7** RankSample for $\mathcal{A}_{\text{CotS}}$

1: **Input:** Model $\theta_T \in \mathbb{R}^p$, Initialisation $\theta_0$,
    canary $\nabla^*$

2: $U \leftarrow \theta_T - \theta_0$
3: **return** $\langle \nabla^*, U \rangle / (\|\nabla^*\|_2 \times \|U\|_2)$

---

**Algorithm 8** Gradient Generation for $\mathcal{A}_{\text{CotS}}$

1: **Input:** dataset $D$, initialization $\theta_0 \in \mathbb{R}^p$,
    clipping threshold $C$

2: $\nabla \leftarrow \mathcal{N}(0, \mathbb{I}_p)$
3: $\nabla^* \leftarrow \nabla / \|\nabla\|_2$
4: **return** $C \times \nabla^*$

---

Figure 9: Algorithms used for auditing via $\mathcal{A}_{\text{CotS}}$: (a) RankSample and (b) Gradient Generation.

We observe experimentally that $\mathcal{A}_{GC}$ outperform $\mathcal{A}_{CotS}$, and that both approaches outperform the loss based adversary $\mathcal{A}_L$, see the results in Figure 7.

**Experiments on auditing with unknown initialization.** As common in differential privacy, we consider throughout the paper that the initialization $\theta_0$. We discuss here the impact of considering the initialization to be unknown. First, we note that standard initialization techniques like Glorot (Glorot and Bengio, 2010) or Xavier (He et al., 2015), which are commonly used to stabilize the training process by preventing the vanishing or exploding gradients phenomena, sample random parameters from a uniform or normal distribution with a small variance. Therefore, such initializations only add a bit of variance. Figure 12 shows the results of using $\mathcal{A}_{GC}$-R to audit a CNN model on CIFAR-10 with unknown initialization from Xavier. We observe no quantitative difference in the performance of $\mathcal{A}_{GC}$-R compared to the case where the initialization is fixed and known.

**Experiments on auditing pre-trained models.** In Figure 12 we present the results of $\mathcal{A}_{GC}$-R to audit pretrained models. We audit an AlexNet model (Krizhevsky et al., 2012), a model pre-trained on ImageNet (treated as public data) and fine-tuned on CIFAR-10 (treated as private data), compared to a CNN trained from scratch directly on CIFAR-10 with no frozen parameters. We see that $\mathcal{A}_{GC}$-R performs tight auditing for $k = 1$ in this scenario as well.

**Privacy profiles.** While in all of our experiments, we report both the theoretical epsilon provided by the privacy accounting and the empirical epsilon at a fixed $\delta = 1e^{-5}$, we can generate the entire privacy profile curve $\varepsilon \mapsto \delta(\varepsilon)$ and thus produce auditing results at an arbitrary $\delta$. Indeed, our auditing technique computes a lower bound on the $\mu$-GDP parameter (see Section A for more details). Using the lower bound on $\mu$ in Corollary 1, we can compute any $\delta(\varepsilon)$. To compute $\varepsilon$ at a fixed $\delta$, we use a line-search algorithm like the bisection method, as the function $\delta(\varepsilon)$ is monotonically increasing in $\varepsilon$. In Figure 11, we present three privacy profile curves $\varepsilon \mapsto \delta(\varepsilon)$ corresponding to the performance of $\mathcal{A}_{GC}$-R when auditing a CNN with $k \in \{1, 5, 25\}$. This confirms that for the case $k = 1$, we are indeed tight in all privacy regimes, while for $k > 1$ $\mathcal{A}_{GC}$-R is not tight for all $\varepsilon \geq 0$, as discussed in Section 5.

**Detailed results for $\mathcal{A}_S^{h^*}$.** In Figure 14 we present all the training curves that generated the heatmap in Figure 3b by computing the empirical privacy loss generated by $\mathcal{A}_S^{h^*}$ for all $\sigma \in \{1, \ldots, 8\}$ and $B \in \{1, 2, 4, 8, 16\}$. As a reference, we display the theoretical epsilon provided by privacy accounting at each $\sigma$. In all plots we observe that the regime $B = 16$ yields tight auditing, and this is also the case for larger batch sizes (not shown to avoid clutter).

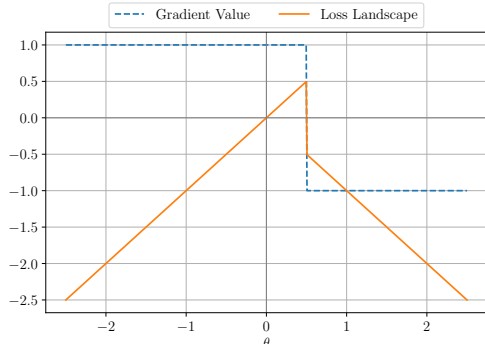

Figure 10: Visualization of the loss landscape corresponding to $A_S^{h^*}$ for $C = 1$.

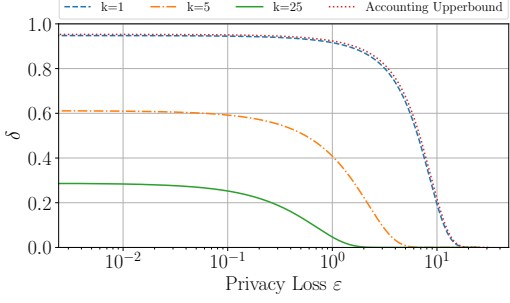

Figure 11: Privacy profiles of $A_{GC}$-R for $k \in \{1, 5, 25\}$, $C = 1$ on CNN trained on CIFAR10 on the last step of training ($T = 250$ for $k = 1$, $T = 1250$ for $k = 5$, $T = 6150$ for $k = 25$).

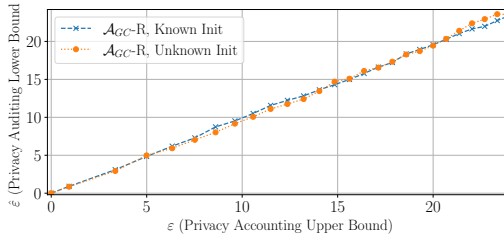

Figure 12: Auditing results for $\mathcal{A}_{GC}$ on CNN trained on CIFAR10 when the initialisation of the neural network is unknown to the adversary, compared to the case of known initialisation.

Figure 13: Auditing results for $\mathcal{A}_{GC}$-R when $k = 1$ on a pre-trained AlexNet model is fine-tuned on CIFAR10.

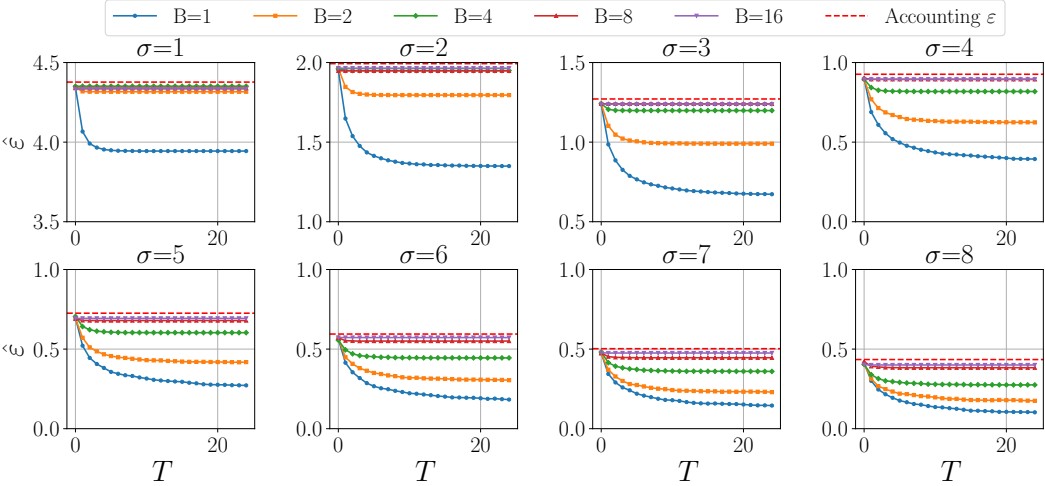

Figure 14: Auditing performance of $\mathcal{A}_S^{h^*}$ over time $T = 25$ for various batch sizes $B \in \{1, 2, 4, 8, 16\}$ and noise levels $\sigma \in \{1 \cdots 8\}$, used to generate Figure 3b.

