# OpenReview forum: "Tighter Privacy Auditing of DP-SGD in the Hidden State Threat Model"
_ICLR.cc/2025/Conference — ICLR 2025 Poster_

### Official Review · Reviewer_fShh · 2024-10-27

**Soundness:** 2
**Presentation:** 3
**Contribution:** 2
**Rating:** 6
**Confidence:** 3

**Summary:**

This paper proposes a privacy auditing method for DP-SGD under a hidden state threat model. Assuming an arbitrary non-convex loss function, the authors abstract away the specific design of neighboring datasets and directly construct gradient sequences for auditing. Their main difference from previous work, which audits DP-SGD with gradient-crafting adversaries at each step, is that they ensures that the gradient sequence is predetermined and not influenced by intermediate outputs, i.e., the sequence is decided offline before training.

The authors considers two types of gradient-crafting:

1. **Random-biased dimension:** randomly selects a dimension and crafts the gradient with the largest possible magnitude in that dimension.

2. **Simulated-biased dimension:** simulates the training algorithm, identifies the least updated dimension, and crafts gradients with the largest possible magnitude there.


Experimentally, they demonstrate that when a canary gradient is inserted at each step, their empirical privacy estimates are tight for common model architectures, including ConvNet, ResNet, and FCNN. The auditing also remains tight when the canary is periodically added with a periodicity of 5. However, at a periodicity of 25, the auditing for some settings may not be tight, suggesting possible privacy amplification by iteration. To investigate this, the authors conducted experiments, showing that their auditing is tight when the batch size and $\epsilon$ are large. This supports prior findings which shows DP-SGD on non-convex problems does not exhibit privacy amplification by iteration. Thus, relaxing the threat model to a hidden state model does not offer better privacy guarantees for certain non-convex problems.

**Strengths:**

1. The paper is clearly written and easy to follow.
2. The proposed method is simple yet empirically effective.
3. The empirical results provide insights into the settings (batch size and $\epsilon$) where DP-SGD may exhibit privacy amplification by iteration for non-convex problems.

**Weaknesses:**

Firstly, the conclusion of this work is not new: the main takeaway—that for some non-convex problems, the hidden state threat model does not lead to better privacy analysis for DP-SGD (i.e., no privacy amplification by iteration)—is the same as the conclusion in previous work [Annamalai, 2024]. However, the methodologies differ: [Annamalai, 2024] constructs a worst-case non-convex loss function for DP-SGD where information from all previous iterates is encoded in the final iterates, while this work directly constructs gradient sequences.

My other concern with this work is that its methodology is also quite limited, though it may be slightly more general than previous approaches (e.g. [Annamalai, 2024]). This work abstracts away the specifics of the loss function and model architecture, focusing directly on gradient crafting. While this simplification aids in designing canary gradients, it resembles a worst-case analysis across all non-convex problems. From a specific gradient sequence, it is not possible to deduce the corresponding loss function and model architecture on which DP-SGD is performed. As a result, the findings do not clarify which specific architectures and loss functions (i.e., types of non-convex problems) may or may not exhibit privacy amplification by iteration.

**Questions:**

The simulated biased dimension method appears quite similar to gradient crafting at each step in terms of implementation [Nasr et al., 2023]. What are the specific differences between these two methods?

---

> ### Author Response · Authors · 2024-11-19
>
> We thank the reviewer for their constructive feedback and address their concerns and questions below.
>
> ## On the contemporaneous work by Annamalai [2024]
> To begin, we want to clarify that, in accordance with the ICLR 2025 Reviewer Guideline, this paper should be regarded as contemporaneous work. Consequently, we are not required to compare our own work to that paper. However, we have chosen to cite and discuss this paper in Remark 4 to provide additional context. In particular, we highlight that our work offers a more nuanced and complete picture than Annamalai [2024], as we uncover the existence of two different regimes and reveal the key role of the batch size.
> We also want to emphasize that the construction presented in Annamalai [2024], like ours, is “synthetic” and it is unclear how to translate it to a practical scenario. We agree that studying how worst-case gradient sequences could be obtained in practical scenarios (i.e., on realistic datasets with commonly used architectures and loss functions) is an important and intriguing direction for future research inspired by our work. Note that, in the context of privacy backdoors, there is exciting progress in achieving arbitrary gradients in practice for specific architectures, see [Feng et al. 2024](https://arxiv.org/pdf/2404.00473).
>
> ## Comparison with Gradient Crafting adversary of Nasr et al. 2023:
> The gradient crafting adversaries of Nasr et al. 2023 (Dirac, Constant, Random) aim to maximize **per-step** privacy leakage. Our simulated biased dimension adversary ($\mathcal{A}_{GC}$-S) is a refined Dirac gradient-crafting adversary that seeks to maximize the **privacy leakage from the final model**. To achieve this, the key difference with the Dirac adversary of Nasr et al. 2023 is that our adversary simulates training to select a favorable dimension (see the detailed presentation in Appendix C, in particular Algorithm 3 therein), rather than selecting the dimension at random as in Nasr et al. 2023. We show that our strategy achieves better privacy auditing for the hidden state model in some scenarios (see Figure 2c).

---

> > ### Comment · Reviewer_fShh · 2024-11-25
> >
> > Thank you for addressing my questions. I have raised my score.

---

### Official Review · Reviewer_xgje · 2024-11-02

**Soundness:** 3
**Presentation:** 3
**Contribution:** 3
**Rating:** 8
**Confidence:** 4

**Summary:**

This paper studies the auditing of last iterate privacy guarantees of DP-SGD for non-convex loss functions. Authors propose an auditing method where a canary gradient is introduced to the DP-SGD steps. Two methods for crafting this "worst-case" canary gradients is proposed: 1. a random direction in the parameter space is selected and a gradient of norm C is inserted to the mnibatch of gradients and 2. the adversary simulates the DP-SGD run and picks as the canary gradient the least updated dimension, again with norm C. Authors demonstrate empirically that if the canary is inserted at every iteration, the auditing bounds correspond to the DP accounting upper bounds, suggesting that the hidden state model does not necessarily provide additional privacy protection. When the canary is inserted less frequently, the adversaries power decreases. Finally, authors propose an auditing setting where the adversary inserts the crafted gradient only in the first iteration and controls the loss landscape. In this setting the auditing bounds match closely the DP bounds when minibatch used in DP-SGD is sufficiently large.

**Strengths:**

Auditing DP algorithms is an interesting and important line of work. Previous work, both in auditing and privacy accounting, have suggested that hiding the intermediate iterations of DP-SGD can be beneficial for the privacy guarantees. In this work, authors improve the privacy auditing of non-convex loss functions, by carefully selecting gradient canaries that get inserted for to the DP-SGD gradient sum. The proposed method significantly improves the existing methods, suggesting that the previous methods have not optimally used the threat model allowed by the hidden state setting.

Authors also study the amplification by iteration effect, by proposing a novel technique that introduces a crafted gradient only for the first step of DP-SGD. The empirical results for this study suggest that there might not be any amplification by iteration for certain non-convex losses, and that only the noise introduced by subsampling the data makes distinguishing the crafted gradient more difficult.

**Weaknesses:**

In general the paper is very well written and the arguments are easy to follow. However, I get a bit confused on the discussion regarding accounting in the section 5.1. Since the threat model studied does not benefit from the subsampling amplification, I don't see any reason of using PRV accounting. When there is no subsampling, the privacy analysis could be performed tightly with Gaussian DP or Analytical Gaussian accounting (Balle et al 2018).

The proposed method assumes that the crafted gradient is possible under the particular loss function. While I do believe that this might be the case for highly over-parametrized models, it would be great if authors could discuss this further in the paper. Would it be possible to somehow trace back the worst-case sample from the worst-case crafted gradient? Similarly the assumption that the adversary can craft the loss landscape to make the crafted gradient the most distinguishable could warrant further discussion.

**Questions:**

- It seems that the privacy accounting upper bounds remain the same for various k (Figures 2 and 3). Now, is this because you have actually accounted only the 250 iterations the canary was inserted? Or do you display the auditing results only up to $T=250$ iteration?
- I don't think there is any reason to use PRV as there is no subsampling. You could just use analytical Gaussian and obtain tight privacy bounds (instead of the upper and lower bounds you get from PRV).
- Caption of Fig4: "Figure 4a gives the ...", I guess you mean 4b?

---

> ### Author Response · Authors · 2024-11-19
>
> We thank the reviewer for their positive feedback and address their questions below.
>
> ## On the usage of PRV accounting
> We have used the default numerical accountant implemented in Opacus out of ease, but we agree that we could just use analytical Gaussian accounting, either via Balle et al. 2018 or through $\mu$-GDP composition. For sanity check, we have implemented this and the results are equivalent to the ones obtained with the numerical accountant. We appreciate the feedback and we will switch to analytical accounting in the final version of the paper.
>
> ## On the craftability of a given sequence of gradients
> DP-SGD, by design, protects any sequence of bounded gradients. Therefore, to assess the tightness of these privacy guarantees (which is our main goal), it makes sense to audit with a worst-case gradient sequence. Interestingly, we are not aware of any theory restricting what gradients can be produced by large neural networks; therefore, one cannot rule out the possibility that there exists some input point that can produce an arbitrary sequence of gradients.
>
> That being said, how to craft such an input point is an interesting open question, with recent work providing some first answers to this question [Feng et al., 2024]. This paper belongs to a recent line of work where the model architecture and parameters are chosen adversarially to leak some input points (a *privacy backdoor*), see the prior work [1, 2, 3] in federated learning. Interestingly, their construction shows that, for a particular choice of architecture/initialisation, a canary point can generate approximately worst-case gradients like the ones used by our gradient-crafting adversaries. In other words, it is sometimes possible to instantiate our adversaries through input-point canaries. However, their work focuses on finding specific instances of models and initializations where this can work and, as far as we know, a general way of crafting arbitrary sequences of gradients has not yet emerged.
>
> [1] Fowl et al. Robbing the Fed: Directly Obtaining Private Data in Federated Learning with Modified Models. ICLR 2022
>
> [2] Boenisch et al. When the Curious Abandon Honesty: Federated Learning Is Not Private. EuroS&P 2023
>
> [3] Boenisch et al. Reconstructing Individual Data Points in Federated Learning Hardened with Differential Privacy and Secure Aggregation. EuroS&P 2023
>
> ## Constant accounting upper bounds
>
> We insert the canary a fixed number of times (250) and only account for those. We have made this more explicit in lines 343-346 of the revised manuscript.
>
> ## On the caption of Figure 4
> Indeed, thanks for catching the typo! We have fixed it in the revised version.

---

> > ### Comment · Reviewer_xgje · 2024-11-25
> >
> > Thanks for addressing my concerns! I'm happy to keep my score as is.

---

### Official Review · Reviewer_aBWh · 2024-11-02

**Soundness:** 4
**Presentation:** 4
**Contribution:** 3
**Rating:** 6
**Confidence:** 3

**Summary:**

This paper proposes a tighter privacy auditing bound by extending gradient-crafting (GC) models to the hidden state regime, where only the last iteration is released. By combining the GC technique with advanced privacy auditing methods, such as privacy auditing using
$f$-DP, the authors achieve a refined privacy auditing bound for the hidden state model.

**Strengths:**

This paper applies GC models to audit various regimes, including both small and over-parameterized models. Additionally, different deep learning architectures (CNN, ResNet, FCNN) are evaluated in the experiments. Overall, the empirical results are convincing.

**Weaknesses:**

The main weakness of this paper is that it does not introduce a new privacy auditing method; rather, it extends the existing gradient-crafting method to the hidden state regime. The primary technical contribution appears to be constructing a sequence of gradients without requiring knowledge of the intermediate gradients at each iteration.

**Questions:**

My question is about the $f$-DP part. In Remark 3, the authors claim that the approximation error from using the central limit theorem (CLT) can be ignored. However, the CLT may underestimate privacy when mixture distributions arise due to shuffling or sub-sampling [A]. For this reason, Nasr et al. (2023) adopt numerical methods, such as FFT, to calculate the privacy profile. For the hidden state model, an
$f$-DP guarantee is provided by [B] without relying on the CLT.

Given these considerations, the authors' assertion that the CLT approximation error can be ignored may not be universally applicable, especially in scenarios involving mixture distributions in (shuffled) DP-SGD. A simple explanation in the main part might be beneficial to the readers.

[A] Unified Enhancement of Privacy Bounds for Mixture Mechanisms via f-Differential Privacy. Wang et al., NeurIPS, 2023.

[B] Shifted Interpolation for Differential Privacy. Bok et al., ICML, 2024.

---

> ### Author Response · Authors · 2024-11-19
>
> We thank the reviewer for their constructive feedback and address their concerns and questions below.
>
> ## On the use of GDP approximation:
>
> Indeed, our Gaussian approximation can underestimate the privacy guarantees when trade-off curves are not symmetric, especially for subsampled or shuffled mechanisms (as underlined by [A]). The main difficulty here is that we do not have a closed-form expression for the Privacy Loss Distribution (PLD) and, in contrast to [Nasr et al. 2023], we cannot use the intermediate steps information to audit a PLD with a known structure. Thus, we use the CLT to approximate the mechanism's output after several optimization steps. As we increase the number of steps in the hidden state, the trade-off function indeed becomes well approximated by a Gaussian trade-off function, as shown in Appendix E. We appreciate the feedback and have added a clarification in Remark 3 of the revised paper along with a reference to [A].
> We note that [B] requires convexity similar to the classic privacy amplification by iteration result. We have cited [B] in the relevant part of the related work section (line 224).

---

> > ### Author Response · Authors · 2024-11-25
> >
> > We thank reviewer aBWh for the provided feedback. As we are approaching the end of the discussion phase, we were wondering if we have addressed the reviewer's concerns. If so, we would greatly appreciate it if they could adjust their score accordingly. Thank you!

---

### Official Review · Reviewer_PjHW · 2024-11-03

**Soundness:** 3
**Presentation:** 3
**Contribution:** 3
**Rating:** 6
**Confidence:** 2

**Summary:**

In the "hidden state" threat model for DP-SGD, an adversary does not have access to intermediate updates and can see only the final model. This paper proposes to audit the hidden state threat model for DP-SGD by introducing adversaries who select a gradient sequence offline (i.e., ahead of when training starts) in order to maximize the privacy loss of the final model even without access to the intermediate models.

**Strengths:**

* The problem setting and results are really interesting and open up new avenues for future work. Identifying different regimes where the gap (between the new auditing lower bounds and the theoretical upper bounds) vanishes and where the gap remains is a nice contribution.
* The paper flows nicely and is and enjoyable to read.
* Simplicity is a virtue and I think the design of the gradient-crafting adversaries for privacy auditing is clever yet also intuitive.

**Weaknesses:**

* The paper uncovers some interesting empirical results but doesn’t offer up much explanation for them. E.g. for regimes where there is a gap, we don’t really get an explanation as to why this gap exists. I do feel that this work falls short of its goal to “enhance our understanding of privacy leakage” (line 537) by reporting the results without interpretation. I think that including more discussions like the “high-level explanation” starting at line 373 would help round out the paper and provide more concrete directions for future work.
* Considering this is one of its main contributions, I found that the description of the adversaries and the privacy auditing scheme in the main paper is vague and not presented with full details.

**Questions:**

* There might not exist a worst-case datapoint that can saturate the gradient clipping threshold at every iteration. Would it be possible to discuss what it means to breach the privacy of a datapoint that does not exist and may in fact be unrealizable?
* It seems like an adversary who could control the sequence of gradients would also be able to infer from that something about the intermediate models. Is this OK for the hidden state threat model?
* Is there any intuition behind privacy amplification only seems to occur for smaller batch sizes?
* Does Implication 1 say anything beyond what Implication 2 says? If the batch size is as large as possible (the entire dataset), then every datapoint will be included in every optimization step of DP-SGD?

---

> ### Author Response · Authors · 2024-11-19
>
> We thank the reviewer for their constructive feedback and address their concerns and questions below.
>
> ##  Explanations for the empirical results:
>
> While we have tried to give some explanations and interpretations for the observed gaps (see lines 414-421 for the gap between $k=1$ and $k>1$ and lines 490-494 for the small batch size regime), we agree with the reviewer that some of these could be expanded.
>
> Here is more intuition regarding the privacy amplification phenomenon occurring at small batch sizes that you specifically ask about. In our worst-case construction, the crafted gradient influences all subsequent gradients computed on genuine points. Recall that gradient updates belong to $[-BC, BC]$ where $B$ is the batch size and $C$ is the clipping threshold. Therefore, as gradient updates are bounded by $[-BC, BC]$ and the noise remains independent of $B$, increasing $B$ allows the adversary to retain more ``signal'' about the canary's presence in subsequent gradients. We have integrated these additional explanations in the revised version of the paper (lines 490-495).
>
> If there are other points where the reviewer would appreciate additional explanations, we would be happy to expand these to clarify our results further.
>
> ## Details on the auditing technique and adversaries:
>
> Regarding auditing, we provide a succinct description in Section 2.2, with more details in Appendix A for the methodology and Appendix D for the experimental details. We believe this is adequate, since we reuse an existing auditing methodology from prior work (Nasr et al., 2023).
>
> On the adversaries, in addition to the description in Section 4, we provide additional details on the implementation of our adversary in Appendix C and more experimental details in Appendix D.
>
> It will not be possible to move these sections into the main body of the paper due to the page limit, but if the reviewer believes it is important to add specific details in the main text, we would be happy to do so.
>
> ## On the existence of a data point yielding a worst-case gradient sequence:
>
> DP-SGD, by design, protects any sequence of bounded gradients. Therefore, to assess the tightness of these privacy guarantees (which is our main goal), it makes sense to audit with a worst-case gradient sequence. Interestingly, we are not aware of any theory restricting what gradients can be produced by large neural networks; therefore, one cannot rule out the possibility that there exists some input point that can produce an arbitrary sequence of gradients.
>
> That being said, how to craft such an input point is an interesting open question, with recent work providing some first answers to this question [Feng et al., 2024]. This paper belongs to a recent line of work where the model architecture and parameters are chosen adversarially to leak some input points (a *privacy backdoor*), see the prior work [1, 2, 3] in federated learning. Interestingly, their construction shows that, for a particular choice of architecture/initialisation, a canary point can generate approximately worst-case gradients like the ones used by our gradient-crafting adversaries. In other words, it is sometimes possible to instantiate our adversaries through input-point canaries. However, their work focuses on finding specific instances of models and initializations where this can work and, as far as we know, a general way of crafting arbitrary sequences of gradients has not yet emerged.
>
> [1] Fowl et al. Robbing the Fed: Directly Obtaining Private Data in Federated Learning with Modified Models. ICLR 2022
>
> [2] Boenisch et al. When the Curious Abandon Honesty: Federated Learning Is Not Private. EuroS&P 2023
>
> [3] Boenisch et al. Reconstructing Individual Data Points in Federated Learning Hardened with Differential Privacy and Secure Aggregation. EuroS&P 2023

---

> > ### Author Response · Authors · 2024-11-19
> >
> > ## On the ability to infer information about intermediate models
> > The essential requirement of the hidden state model is that the adversary observes only the final model. Thus, as long as adversaries do not exploit the knowledge of intermediate models (our gradient-crafting adversaries do not), they are valid for this threat model. Observing the final model certainly reveals some information about intermediate steps, but ultimately what matters is how much information is revealed about the presence or absence of the canary.
> >
> > ## Clarification for Implication 2
> >
> > We thank the reviewer for this question. Implication 1 states that when the canary is inserted at every step ($k=1$), the bounds are tight. On the other hand, Implications 2 and 3, which are intended to be interpreted together, describe the two different regimes that emerge when $k>1$: the large batch regime (Implication 2) and the small batch regime (Implication 3). Implication 2 highlights that when $B/\sigma$ is sufficiently large, the accounting remains tight even for $k>1$. An example illustrating a scenario that falls within the scope of Implication 1 but not Implication 2 is when the dataset size $|D|=1$, $B=1$ and $\sigma$ is large. We have slightly modified the text before Implications 2-3 in the revised version to make this more clear.

---

> > > ### Author Response · Authors · 2024-11-25
> > >
> > > We thank reviewer PjHW for the provided feedback. As we are approaching the end of the discussion phase, we were wondering if we have addressed the reviewer's concerns. If so, we would greatly appreciate it if they could adjust their score accordingly. Thank you!

---

> > > > ### Comment · Reviewer_PjHW · 2024-11-25
> > > >
> > > > Thanks for the detailed response, and for the additional text explaining the small batch size phenomena. I have raised my score accordingly.
> > > >
> > > > Re: existence of a datapoint yielding the worst-case gradient sequence. This is certainly an interesting conundrum, as there is neither proof it exists or that it doesn’t exist. I do think that it would be useful to include a caveat making this point clear.
> > > >
> > > > And — I agree page limitations won’t allow for including the entire algorithm blocks describing the two adversaries — but I wasn’t sure I fully grasped how the adversaries fit together with the auditing scheme. Algorithm 1 takes as input a canary point $x^*$, and it looks like RankSample would have to be adapted for the case of gradient-crafting adversaries where there is no explicit datapoint. So it would be great to see more details about how these gradient-crafting adversaries could plug into an algorithm like Algorithm 1.

---

> ### Author Response · Authors · 2024-12-04
>
> We thank the reviewer for their positive feedback and relevant suggestions!
>
> We will clarify in Section 4 that, to our knowledge, there is no restriction on the gradient sequences generated by a neural network. We also agree that it would be useful to explicitly integrate the concrete auditing algorithm for our proposed adversaries in the generic template described by Algorithm 1.
>
> We will incorporate these updates in the final version of our manuscript, as we cannot upload a revised version during this part of the discussion phase.

---

### Meta-Review · Area_Chair_zxEB · 2024-12-20

**Metareview:**

The paper examines the extent to which not revealing the intermediate states provides additional privacy for training ML models. Prior work has shown "privacy amplification via iteration" when intermediate states are not revealed, for convex objectives. The paper shows that when an adversarially crafted gradient is inserted at every step there is not additional privacy in only releasing the final model. If a crafted gradient is not inserted at every step and the batch size is small then privacy amplification seems possible. The paper suggests several questions for future work, and I encourage the authors to try and distill down some of these questions and potential explanations in the final version of the paper (as they started to do during the author response).

**Additional Comments On Reviewer Discussion:**

The reviewers raised some questions which were addressed by the authors, and have been accounted for by the reviewers:

1. Explanations for the empirical results
2. Comparison with contemporaneous work
3. Issues with writing

---

### Decision · Program_Chairs · 2025-01-22

Accept (Poster)